

# The dynamics of the carbon dioxide system in the outer shelf and slope of the Eurasian Arctic Ocean

Irina I. Pipko[1,2], Svetlana P. Pugach[1,2], Igor P. Semiletov[1,2,3], Leif G. Anderson[4], Natalia E. Shakhova[2,3], Örjan Gustafsson[5,6], Irina A. Repina[7], Eduard A. Spivak[1], Alexander N. Charkin[1,2], Anatoly N. Salyuk[1,2], Kseniia P. Shcherbakova[1,2], Elena V. Panova[2], Oleg V. Dudarev[1,2]

[1]V.I. Il'ichev Pacific Oceanological Institute, Russian Academy of Sciences, Vladivostok, 690041, Russia
[2]National Research Tomsk Polytechnic University, Tomsk, 634050, Russia
[3]International Arctic Research Center, University Alaska Fairbanks, Fairbanks, AK 99775, USA
[4]Department of Marine Sciences, University of Gothenburg, Gothenburg, 412 96, Sweden
[5]Department of Environmental Science and Analytical Chemistry, Stockholm University, Stockholm, 10691, Sweden
[6]Bolin Centre for Climate Research, Stockholm University, Stockholm, 10691, Sweden
[7]A.M. Obukhov Institute of Atmospheric Physics, Russian Academy of Sciences, Moscow, 119017, Russia

*Correspondence to*: Irina I. Pipko (irina@poi.dvo.ru)

**Abstract.** The Arctic now is undergoing dramatic changes, which cover the entire range of natural processes; from extreme increases in the temperatures of air, soil, and water, to changes in the cryosphere, the biodiversity of Arctic waters, and land vegetation. Small changes in the largest marine carbon pool, the dissolved inorganic carbon pool, can have profound impact on the carbon dioxide ($CO_2$) flux between the ocean and the atmosphere, and the feedback of this flux to climate. Knowledge of relevant processes in the Arctic seas improves the evaluation and projection of the carbon cycle dynamics under conditions of rapid climate change.

Investigation of the $CO_2$ system in the outer shelf and continental slope waters of the Eurasian Arctic seas (the Barents, Kara, Laptev, and East Siberian seas) during 2006, 2007, and 2009 revealed a general trend in the surface water $pCO_2$ distribution, which manifested as an increase in $pCO_2$ values eastward. Existence of this trend was determined by different oceanographic and biogeochemical regimes in the western and eastern parts of the study area; the trend is likely increasing due to a combination of factors determined by contemporary change in the Arctic climate, each change in turn evoked a series of synergistic effects. A high-resolution in situ investigation of the carbonate system parameters of the four Arctic seas was carried out in the warm season of 2007, which was characterized by the next-to-lowest historic sea ice extent in the Arctic Ocean to that date. The study showed the different responses of the seawater carbonate system to the environment changes in the western vs. the eastern Eurasian Arctic seas. The large open, highly-productive water area in the northern Barents Sea enhances atmospheric $CO_2$ uptake. In contrast, a growing $CO_2$ evasion occurs in the outer shelf and slope waters of the East Siberian Arctic seas as a result of the increasing influence of river runoff and degradation of terrestrial organic matter, in combination with the high surface-water temperature due to the warm air temperature and decreasing albedo during sea ice free conditions.





This investigation shows the importance of processes that vary on small scales, both in time and space, for estimating the air-
sea exchange of $CO_2$. It stresses the need for high-resolution coverage of ocean observations as well as time series.
Furthermore, time series must include multi-year studies in the dynamic regions of the Arctic Ocean during these times of
environmental change.

# 1 Introduction

The Arctic now is undergoing dramatic changes, which cover the entire range of natural processes; from extreme increase in
the temperatures of air, soil, and water, to changes in biodiversity of Arctic waters and land vegetation (Serreze and Barry,
2011; Bhatt et al., 2010). In 1896, the Swedish scientist Svante Arrhenius hypothesized that changes in the atmospheric
concentration of carbon dioxide ($CO_2$) could alter the earth's surface temperature and that this temperature change would be
especially large in polar latitudes. This likely is the first formal description of what today is known as the Arctic
amplification, i.e., a higher temperature increase in Arctic regions than in other regions of the globe (Serreze and Barry,
2011; Jeffries et al., 2013). The changes being observed today will probably become more intense in the coming decades
through positive feedback, causing further changes in atmospheric circulation, river discharge, carbon cycle, conditions of
terrestrial and submarine permafrost, vegetation, and many other natural processes; the consequences will be noticed within,
as well as outside of the Arctic region (Serreze and Barry, 2011; Anderson et al., 1998; Macdonald et al., 2008; Semiletov et
al., 2000, 2016; Shakhova et al., 2009, 2014). Currently, these changes refer to a "new condition" of the Arctic climate
(Kattsov et al., 2010; Jeffries et al., 2013; Wood et al., 2015).

The most obvious indicator of Arctic climatic change is the change of sea ice cover, with a persistent decline in areal extent
during the last decades. Since the start of satellite observations in 1979 the sea ice extent in March, the period of maximum
coverage, has declined by 2.6 % per decade (Serreze et al., 2007; Stroeve et al., 2012; Jeffries et al., 2013). However, during
the last decade the September sea ice extent has decreased by 13 % relative to the 1979 – 2000 average (Jeffries et al., 2013).
The change in sea ice coverage is most pronounced in the large Eastern Arctic shelf seas (ftp://sidads.colorado.edu).
Moreover, the melt season has lengthened by 1-2 weeks per decade, and with continued Arctic warming it will expand
further (Stroeve et al., 2014).

The environmental conditions vary between the Eurasian shelf seas, of which the Barents Sea is one of the largest and
deepest (Jakobsson, 2002). According to the classification of Carmack et al. (2006), the Barents is considered an "*inflow*"
shelf sea or an Atlantic-influenced shelf sea (Findlay et al., 2015). The general inflow of warm and salty water from the
Atlantic keeps a large part of the Barents Sea ice-free all year around. In the northern area a portion of the warm Spitsbergen
Current returns around Svalbard to the Barents Sea as "cold Atlantic" water (Kaltin et al., 2002) or "Arctic" water (Loeng,
1991). This area is largely covered with ice, but at a variable extent over the year, and the presence of polynyas contributes
to salinization of water in the winter (Carmack et al., 2006). However, the cooling of the Atlantic water (AW) during its
passage through the Barents Sea produces the largest volume of high-density water that ventilates the deep Eurasian Basin





water (Schauer et al., 2002). The Barents Sea receives little river input compared to other Arctic shelf seas (Anderson et al., 1998; Schauer et al, 2002). Due to high primary productivity (PP) and cooling during transit to the north, the waters of the Barents Sea constitute the strongest all-season sink for atmospheric $CO_2$ in the Arctic (Fransson et al., 2001; Omar et al., 2007; Bates and Mathis, 2009; Årthun et al., 2012; Lauvset et al., 2013).

The other shelf seas (the Kara, Laptev, and East Siberian seas) are classified as "*interior*" shelf seas (Carmack et al., 2006) or river-influenced shelf seas (Findlay et al., 2015). The river discharge as well as the seasonal formation and melting of sea ice greatly impacts the hydrology and chemistry of these shelf seas. Water of Atlantic origin enters the Kara Sea from the Barents Sea, and the Kara Sea also receives more than a third of the volume of riverine discharge flowing into the Arctic Ocean, mainly by the Ob and Yenisei rivers.

The shallow East Siberian and Laptev seas, together with the Chukchi Sea, form a large, broad, and shallow province composing as much as 22 % of the entire Arctic Ocean area but only 1 % of the volume (Jakobsson, 2002). The Laptev Sea and the East Siberian Sea are surrounded and underlain by permafrost and are characterized by the degradation of coastal ice-complex and terrestrial permafrost containing an extensive pool of ancient labile organic matter (OM) (Semiletov, 1999; Sánchez-García et al., 2014; Schirrmeister et al., 2011; Tesi et al., 2014; Vonk et al., 2014). They are strongly impacted by

the input and transformation of terrestrial OM (Charkin et al., 2015; Dudarev et al., 2006; Gustafsson et al., 2011; Semiletov et al., 2007, 2016; Tesi et al., 2016; Bröder et al., 2016; Vonk et al., 2014). The input of suspended as well as dissolved terrigenous material (Alling et al., 2010; Raymond et al., 2007; Holmes et al., 2012) including optically-active fractions of dissolved organic matter, colored dissolved organic matter, CDOM (Pugach et al., 2015) and the presence of ice cover throughout a significant part of the year reduces the depth of solar radiation penetration into the water column. Together with

limited nutrient content, these conditions make these seas unproductive, 5-10 times less productive than inflow shelves (Carmack et al., 2006). Moreover, intense inflow of terrigenous OM, low productivity, as well as subsea release of methane that partly oxidizes (Shakhova et al., 2015), make significant areas of these seas heterotrophic; $CO_2$ sources to the atmosphere (Anderson et al., 2009, 2011; Pipko et al., 2005, 2011a; Semiletov et al., 2007, 2012, 2013).

The waters of the Arctic seas have become warmer and fresher than they were several decades ago (Wood et al., 2015);

among other effects, this warming and freshening has added increased OM input to the shelf seas (Semiletov et al., 2016). How climate change impacts the contemporary carbon cycle in the Eurasian Arctic Seas, including its consequences for transformation and fluxes, has been the subject of intense interest during the last decade (Anderson et al., 2011; Bischoff et al., 2016; Bröder et al., 2016; Charkin et al., 2015; Gustafsson et al., 2011; Karlsson et al., 2016; Macdonald et al., 2008; Sánchez-García et al., 2014; Tesi et al., 2014; Vonk et al., 2014). Small changes in the largest marine carbon pool, the

dissolved inorganic carbon (DIC) pool, can have profound impact on the $CO_2$ flux between the ocean and the atmosphere, and the feedback of this flux to climate. Knowledge of relevant processes in the Arctic seas improves the evaluation and projection of the carbon cycle dynamics under conditions of rapid climate change.

The East Siberian Arctic seas (ESAS), including the Laptev Sea, the East Siberian Sea, and the Russian sector of the Chukchi Sea, are especially relevant in this perspective because they have the broadest and the shallowest shelves among the



Arctic seas, they receive large volumes of river discharge, they are characterized by high rates of coastal erosion, and their drainage basins are underlain by permafrost (Macdonald et al., 2008; Semiletov et al., 2000, 2012).

Studies of the carbonate system of the Barents Sea (Årthun et al., 2012; Lauvset et al., 2013; Pipko et al., 2011b; Yakushev and Sørensen, 2013) as well as the ESAS waters (Anderson et al., 2009, 2011; Pipko et al., 2011a, 2015, 2016; Semiletov et al., 2012, 2013, 2016) have been performed during the last decade. The Kara Sea remains less studied in the context of carbonate system dynamics and the main research was accomplished in the shallow part of this sea (Makkaveev et al., 2010, 2015). Most of the ESAS studies also were carried out in ice-free areas, i.e. at depths normally limited to 70 m isobaths, which corresponds to the area of the inner and middle shelves. Meanwhile, the deep part of the seas where the changes in the ice cover are most pronounced is the least investigated. To date, only one paper dedicated to the dynamics of the $CO_2$ system in East Siberian Sea outer shelf is available (Anderson et al., 2017); it is based on the field campaign accomplished within the framework of the international SWERUS project onboard the Swedish ice-breaker Oden. This study was accomplished in ice conditions, with the sea ice concentration ranging between 70 and 100 %.

The objective of this contribution is to evaluate the importance of the meteorological and oceanographic conditions and biochemical processes, which determine the surface water partial pressure of $CO_2$ ($pCO_2$) and air-sea $CO_2$ fluxes in the outer regions of the Eurasian Arctic seas shelf/slope system along the AW inflow path in different years. This will add to the knowledge of the regional sensitivity to current changes and thus project the response of the entire Arctic carbon cycle to global climate warming.

## 2 Materials and methods

### 2.1 Study area

The study is based on observational data collected in the Eurasian sector of the Arctic Ocean during the summer-fall (late August-early October) seasons of 2006, 2007, and 2009 (Fig. 1). An extensive investigation of the outer shelf and continental slope of the Barents, Kara, and Laptev seas and the northeastern East Siberian Sea was performed during the 2007 expedition on the research vessel "Viktor Buynistkiy" within the framework of the Nansen and Amundsen Basins Observational System (NABOS) program (Fig. 1). Data collected in 2006 and 2009 within the framework of the NABOS program on board the icebreaker "Kapitan Dranitsyn" were used for comparative analysis (Fig. 1).

### 2.2 Methods

#### 2.2.1 Hydrological data

During all cruises, water samples for chemical analysis were taken with a standard Rosette system equipped with the SBE19+ CTD (conductivity, temperature, depth) probe to record conductivity and temperature. In 2007 another SBE19+





probe equipped with the same sensors was deployed in a 150 L plastic barrel into which flowing seawater was pumped from
the depth of ~4 m at the rate of about 80 liters per minute.

### 2.2.2 Total alkalinity ($A_T$)

Water samples were poisoned with a mercuric chloride solution at the time of sampling to halt biological activity (Dickson et
al., 2007) and were stored in the dark at room temperature until they were analyzed ashore. $A_T$ measurements were
performed within one month of sampling using an indicator titration method according to Bruevich (1944) with a precision
of ~2 μmol kg$^{-1}$ with the accuracy set by calibration against certified reference materials (CRMs) supplied by A. Dickson,
Scripps Institution of Oceanography (USA).

### 2.2.3 pH

A potentiometric method was applied to determine pH using a cell without a liquid junction (Tishchenko et al., 2001, 2011)
and reported on the total hydrogen ion concentration scale (Dickson et al., 2007). The precision of pH measurements was
about 0.004 pH units.

### 2.2.4 Partial pressure of carbon dioxide ($pCO_2$)

Continuous measurements of $pCO_2$ were performed in the surface mixed layer using a Submersible Autonomous Moored
Instrument for $CO_2$ (SAMI-$CO_2$ Sensor) with a precision of ±1 μatm (DeGrandpre et al., 1995). The sensor was deployed in
the same barrel as the SBE19+ probe. The calibration procedures are described in detail in DeGrandpre et al. (1995). The
temperature in the barrel was 0.55 °C higher than the sea-surface temperature and the $pCO_2$ measurements were corrected to
in situ temperature using the equation of Takahashi et al. (1993). All in situ surface data described in this paper were
averaged over 30-minutes intervals as was done for the shallow Arctic Eurasian seas (Semiletov and Pipko, 2007).
At oceanographic stations surface $pCO_2$ values were calculated from pH and $A_T$ using the CO2SYS program of Lewis and
Wallace (1998) with equilibrium constants of Mehrbach et al. (1973) refit by Dickson and Millero (1987).

### 2.2.5 Wind speed

The wind speed was measured using an automated meteorological station (Gradient Automatic Weather Station AWS2700)
located at a height of 15-20 m above sea level. The true wind speed was calculated using navigation information and was
extrapolated to the height of 10 m.

### 2.2.6 $CO_2$ flux calculation

Air-sea $CO_2$ fluxes were calculated using the diffusive boundary layer model:

$$F = K_w\ S\ \Delta pCO_2\ (1-f_{ice}), \qquad\qquad\qquad (1)$$





where F is gas flux (e.g. in mmol $CO_2$ m$^{-2}$ day$^{-1}$), $K_w$ is gas-transfer velocity, S is $CO_2$ solubility (Weiss, 1974), $\Delta pCO_2$ is the difference between the atmospheric and oceanic $pCO_2$, and $f_{ice}$ is the fraction of sea ice coverage. Two relationships were used for calculating gas-transfer velocity (Wanninkhof, 1992 and Wanninkhof and MacGillis, 1999). Gas-transfer velocities were calculated using onboard measured wind speed.

### 2.2.7 Apportionment of freshwater (FW) fractions

In order to determine the composition of water samples, we used a three-component mass balance, using salinity and $A_T$ in this evaluation (e.g. Ekwurzel et al., 2001; Fransson et al., 2001, 2009). The major FW sources are river water (RW) and sea ice meltwater (MW), both mainly originating from the Arctic shelf areas. It is assumed that each summer sample is a mixture of Atlantic-derived seawater ($f_{SW}$), river water ($f_{RW}$), and sea ice meltwater ($f_{MW}$). For riverine $A_T$, the average value of 840 µmol kg$^{-1}$ was applied, which is typical of the largest Siberian rivers (the Ob, Yenisei, and Lena rivers) during the warm season (Tank et al., 2012); up to 90 % of the total river discharge enters the Arctic Seas during this season (Dittmar and Kattner, 2003). The sea ice values of salinity and $A_T$ are taken from Fransson et al. (2009); the Atlantic-derived water values of salinity and $A_T$ are taken from Pipko et al. (2011b).

This gives us the following equations for computing the mass balance:

$$1 = f_{SW} + f_{RW} + f_{MW}; \tag{2}$$

$$S = 34.90\ f_{SW} + 5\ f_{MW}; \tag{3}$$

$$A_T = 2292\ f_{SW} + 840\ f_{RW} + 349\ f_{MW}. \tag{4}$$

### 2.2.8 Statistical treatment and graphical representation of the data

Data were tested statistically using an empirical distribution function test in the Statistics 7.0 software package. Descriptive statistics were calculated for the 95 % confidence interval of the mean (P = 0.95, alpha = 0.05). Most of the plots and maps in this study were created with the Ocean Data View software (Schlitzer, 2011).

## 3 Results and Discussion

### 3.1 Meteorological conditions

*In the summer season of 2006*, low sea level pressure (SLP) dominated over the Arctic Ocean (Fig. 2a). It resulted in dominating westerly winds that hampered penetration of RW into the central Arctic Ocean, as well as northern to northwestern sea ice drift. The area of sea ice cover was maximal (5.9 million km$^2$) and the sea ice edge in 2006 had the most southern position of the three years studied (Fig. 1a), which also impeded the transfer of surface water to the deeper part of the ocean. The negative sea ice concentration anomaly was strongest to the east of the study area, while the total





anomaly for the whole Arctic Ocean was -0.5 million km$^2$ compared to the mean coverage during the 1981-2010 time period
(ftp://sidads.colorado.edu). The sea ice conditions varied from light to ice free in the central Laptev Sea to heavy north of the
Novaya Zemlya islands where the concentration reached 90 – 100 %.

The Arctic Dipole (AD), which is characterized by low SLP on the Eurasian side of the Arctic and high SLP on the
American side, was present *in the summer of 2007* (Fig. 2b) and contributed to a 2007 record minimum sea ice extent

(Overland et al., 2014). The AD pattern persisted for part of the summer during each year following 2007. The interplay
between the two regional centers of atmospheric pressure controlled the wind pattern, especially over the ESAS. The
maximum summer winds and the most intensive transfer of RW and sea ice to the north and northwest occurred in the warm
season of 2007, forced by the extreme pressure gradient between the two centers of action (Fig. 2b). In fact, strong winds
were experienced during the 2007 cruise, with wind speed reaching 22 m sec$^{-1}$ in the western study area and 13 m sec$^{-1}$ in the

eastern. In 2007 the sea ice anomaly reached -1.6 million km$^2$ for the entire Arctic Ocean. The negative anomaly in the sea
ice concentration shows how much the ice concentration for a month differs from the mean calculated for that month over
the 1981 through 2010 time range; that mean was maximal in the ESAS, exceeding 50 % (ftp://sidads.colorado.edu).
Actually, the entire study region was largely ice-free in 2007.

A high-pressure area was also present over the American side of the Arctic Ocean *in the summer of 2009*, with comparable

pressure in the center of the anticyclone but with an even larger extent than in 2007 (Fig. 2c). However, the low-pressure
center over the Siberian Arctic was much weaker, resulting in a weaker SLP gradient and correspondingly weaker wind. A
significant part of the study area was covered with ice in 2009, with the sea ice concentration reaching 95-100 % (Fig. 1).
The total Arctic Ocean anomaly of sea ice concentration in 2009 was also negative at -1.0 million km$^2$
(ftp://sidads.colorado.edu). Maximal interannual deviations in the sea ice extent were detected in the ESAS, while the

position of the sea ice edge in the Barents Sea was far to the north and did not differ much between the years (Fig. 1a).

### 3.2 Oceanographic observations

### 3.2.1 Temperature

In 2006, the temperature was close to the freezing point in the regions where sea ice is present, in the northern Barents and
Kara seas as well as in the deep basin north of the Laptev Sea; surface temperatures varied from -1.82 to 3.30 ºC. The

highest temperatures (maximum 3.90 ºC) were found in the ice-free waters of the Laptev Sea (Fig. 3). In 2007, low sea
surface temperatures were found in the northern Barents Sea east of Svalbard, in the northern Kara Sea, and in Vilkitsky
Strait (down to -1.13 ºC, Fig. 4). These low temperatures were associated with the ice edge vicinity and the presence of sea
ice in Vilkitsky Strait (Fig. 1). The waters in the northwestern Barents Sea retained the original AW characteristics (2<T<5
and S>34.8, Hopkins, 1991). The highest sea surface temperatures (up to 4.11 °C) were measured west and north of Svalbard

(Fig. 4). High surface water temperatures were also measured in the Laptev Sea (~ 3.70 ºC) and over the Lomonosov Ridge





(Fig. 4). In 2009, surface water temperatures varied from -1.73 to 2.85 ℃ (Fig. 5). Temperature remained < 0 °C over most of the study area with > 0 °C values in the ice-free areas of the Laptev Sea and the Kara Sea.

### 3.2.2 Salinity

In 2006, the sea surface salinity, as measured at the oceanographic stations, covered a range from 27.00 to 33.57 (Fig. 3). High salinity was observed along the AW inflow path, i.e. in the northern Barents and Kara seas. As it enters the northern Laptev Sea the water's salinity decreases through mixing with river runoff; the lowest salinity was measured in the ice-free waters of the Laptev Sea (Fig. 3). The sea surface salinity within the study region in 2007 varied substantially, as was expected considering the differences in oceanographic regimes of the studied seas. In the West Spitsbergen Current, the salinity of the AW was close to 35; it slowly decreased along the cruise track in the northwestern Barents Sea, slightly varied in the central part of the sea, and decreased in the eastern part (Fig. 4). In the eastern Kara Sea, which is influenced by river runoff, the salinity decreased to well below 30, a salinity level that also was observed in the western Laptev Sea as well as at the Laptev Sea continental slope. The lowest salinities were observed in the eastern Laptev Sea and northwestern East Siberian Sea, reaching well below 25 (Fig. 4). In 2009, the surface water salinity varied from 28.79 to 33.88 with relatively constant salinity in the waters of the Barents and the Kara seas, while it dropped sharply when entering the Laptev Sea where the lowest values were observed (Fig. 5).

### 3.3 Surface water pCO$_2$ spatial distribution in 2007

Seawater pCO$_2$ is affected by several processes. Some are physical, such as temperature and vertical as well as horizontal advection; others are biological, such as production/mineralization of OM. The importance of these processes for the surface pCO$_2$ values observed in the fall season of 2007, which was characterized by the next-to-lowest historic sea ice extent in the Arctic Ocean to that date, is discussed in the following.

### 3.3.1 The Barents Sea

The Barents Sea is an inflow shelf where the supply of nutrients from the Atlantic forms the basis for high PP. This, together with the accompanying heat loss, results in year-round CO$_2$ undersaturation of the surface layer. Consequently it is an annual sink for atmospheric CO$_2$, though with large spatial variability (Fransson et al., 2001; Kaltin et al., 2002; Nakaoka et al., 2006; Omar et al., 2007; Bates and Mathis, 2009; Lauvset et al., 2013). Of the Arctic shelf seas, it can only compare with the highly productive Chukchi Sea where the surface pCO$_2$ can drop down to 100 µatm during the warm productive season (Bates, 2006; Pipko et al., 2002).

Our investigation shows that the surface waters in the less-studied northern Barents Sea were also undersaturated and thus the northern Barents Sea is a sink of atmospheric CO$_2$ (Fig. 4). The waters to the west of Svalbard are undersaturated by about 50 µatm, which is typical for this region when the flux from the atmosphere cannot keep up with the decrease in pCO$_2$ caused by the cooling of the northward-flowing water. During its flow to the north/northeast the surface water is cooled





further, mainly by melting sea ice; thus, it also freshens. Therefore, the temperature of surface waters decreased by ~4.5 ºC (from ~4 ºC to ~-0.5 ºC, Fig. 4) at 9-24 ºE longitude; this caused a thermodynamic decrease of $pCO_2$, calculated according Takahashi et al. (1993), by ~70 µatm. The correlation between temperature and $pCO_2$ in this region was strong (R = 0.84),

further emphasizing the importance of temperature in determining $pCO_2$ here (Fig. 6). An additional $pCO_2$ decline resulted from the addition of sea ice MW, normally undersaturated in $CO_2$ (Nedashkovsky and Shvetsova, 2010; Rysgaard et al., 2012); the calculated sea ice MW fraction was up to 8-10% in the northeastern Barents Sea (Fig. 7). PP could have also contributed to lowering $pCO_2$ even if this study occurred at the end of the productive season (end of September – beginning of October) because air – sea exchange might not have compensated fully for the biological drawdown. The remaining effect

from the late PP can be illustrated by water supersaturation in dissolved oxygen (up to 113 % of saturation) down to a depth of ~25 m, measured in September 2006 and September 2009.

To the east, in the northern Barents Sea, surface temperatures slightly varied with longitude and remained negative; salinity slowly decreased from 24-53 ºE longitude, and then reduced to ~ 31 near 65º E longitude (Fig. 6). In the eastern Barents Sea, the relationship of $pCO_2$ values and temperature was weak; salinity demonstrated a significant spatial variability, but a

$pCO_2$-salinity correlation was practically absent (Fig. 6). Therefore, in the northern Barents Sea, $pCO_2$ variability is driven by different processes in northeastern and northwestern parts of the sea; the temperature impact predominates in the west, and the influence of MW predominates eastward.

### 3.3.2 The Kara Sea

The Kara Sea surface waters were undersaturated in $pCO_2$ with the lowest values in the west and highest to the east (Fig. 4).

The $pCO_2$ correlation with temperature was weak (R = 0.28) for the entire Kara Sea, but $pCO_2$ was strongly negatively correlated with salinity (R = -0.72). Generally, the Kara Sea has two oceanographic sub-regions with different regimes, the *western part* where Atlantic origin water dominates, slightly modified by sea ice melt, and the *eastern part* where this modified water is further diluted by river runoff from the two Great Siberian rivers, the Ob and the Yenisei. This pattern is set by the general eastward direction of water transport in the Eurasian Arctic seas (e.g., Olsson and Anderson, 1997), but

wind is the main driving force of the north RW flow in the Kara Sea (Harms and Karcher, 1999). In early summer 2007, with an intensive development of the AD (Fig. 2), a western transport of RW was developed; in late summer a northern/northeastern type of RW distribution predominated (Zatsepin et al., 2010) and the deep part of the sea was partly influenced by these waters. Note that in 2007 the discharge of the Ob and Yenisei rivers was the largest of the three years studied (2006, 2007, and 2009) and exceeded (+23 % and +4 %, respectively) the average multi-year value for the 1999-

2009 period (426 and 663 km³, respectively) (PARTNERS and ArcticGRO Projects data); this river discharge additionally affected the northeastern Kara Sea.

Hence, we examine separately the relationships of $pCO_2$ with the hydrographic characteristics of the western and eastern regions (Fig. 8). The lack of reliable correlation of $pCO_2$ with the hydrography in the western Kara Sea was analogous to the northeastern Barents Sea. This emphasizes the fact that similar source of waters as well as processes that determine the





carbonate system dynamics occurred in this part of the Eurasian Arctic seas; sea surface temperature slightly changed and MW was the predominant source of fresh water.

In contrast, processes determining carbonate system dynamics in the eastern Kara Sea were clearly different. The examination revealed a highly positive correlation between $pCO_2$ and temperature (R = 0.84) and a strong negative correlation of $pCO_2$ with salinity (R = -0.62) in the eastern part of the sea (Fig. 8). Together with the increase of $pCO_2$

toward the east, significant correlations of $pCO_2$ values with hydrological parameters pointed to the role of RW. Riverine discharge added to the seawater warming, increased $pCO_2$, and contributed water enriched with $CO_2$; $pCO_2$ was also increased via decay of terrestrial OM, transported with RW.

The negative correlation found in the eastern Kara Sea is typical for regions influenced by warm RW, with high $pCO_2$ and labile OM as a substrate for further $CO_2$ production (Semiletov et al., 2013). Furthermore, together with salinity, $pCO_2$ is a

useful tracer of the river plume distribution within the Kara Sea (Fig. 8).

Note that, despite the presence of RW in the eastern part of the sea, the surface $pCO_2$ values remained below atmospheric values.

### 3.3.3 The Laptev Sea

High $pCO_2$ was observed in the Laptev Sea surface waters, even to levels that exceeded atmospheric (Fig. 4).

Supersaturation was observed in the southern Laptev Sea outside the Lena River Delta, which is typical in surface waters of the eastern Laptev Sea inner and middle shelves (Anderson et al., 2009; Semiletov et al., 2013; Pipko et al., 2016). Moreover, high surface water $pCO_2$ was also observed in the surface waters of the outer Laptev Sea shelf over the Lomonosov Ridge and further to the east (Fig. 4). The computed RW fraction reached 30-35 % in the southern Laptev Sea and decreased to 5-7 % north of the latitudinal transect extending from the Lena Delta (Fig. 7). High RW content in the

surface water (up to 33 %) was also found over the Lomonosov Ridge, which indicated a more efficient northeastern transport of RW in the ice-free conditions of 2007. Thus, under intensive AD development the northeastern transfer of RW prevailed and maximal RW content was found in the surface slope water over the Lomonosov Ridge. The MW played a small role on the middle and outer shelves; a strong sea-ice-related brine signal was found here (the brine fraction reached 10 %). In the western outer shelf of the Laptev Sea the surface waters were undersaturated in $CO_2$ relative to atmosphere; the

lowest values were observed furthest to the north over the deep basin. This was the region where Atlantic origin water mixed with MW dominated (the MW fraction increased by up to 10-15 %); the temperature was low, as was the fraction of RW (Figs. 4, 7).

Similar to the eastern Kara Sea, strong correlations were found between $pCO_2$ and temperature/salinity in the Laptev Sea, the recipient of the large Lena River inflow. The correlation was positive with temperature (R = 0.78) and negative with salinity

(R = -0.59). The surface water salinity of the Laptev Sea was lower than that of the Kara Sea (Fig. 4) with the FW source dominated by riverine runoff (Fig. 7). Thus, higher $pCO_2$ values were found in the Laptev Sea than in the Kara Sea, including levels of supersaturation.



Consequently, large areas of $CO_2$ out-gassing to the atmosphere were identified in the Laptev Sea, areas that might increase their strength as the climate warms. This is because when the permafrost thaws, more terrestrial OM will be exposed to
microbial mineralization, producing $CO_2$ both within the drainage basins and in the river plume within the shelf sea. The effect is strengthened by increased river discharge and by coastal erosion as a result of increasing water temperature and intensified wind and wave activity when the sea ice cover decreases (Serreze et al., 2007; Shakhova et al., 2014, 2015). Moreover, the supply of large quantities of optically-active dissolved OM and suspended material (Dudarev et al., 2006; Sánchez-García et al., 2014; Vonk et al., 2014; Pugach et al., 2015; Charkin et al., 2015) promotes the accumulation of solar
radiation in the surface layer, which increases the heat content leading to further lengthening of the sea-ice-free season. We also suggest that progression of subsea permafrost thawing and decrease in ice extent could result in a significant increase in carbon discharge from the sea floor (Nicolsky and Shakhova, 2010; Shakhova et al., 2014, 2015; Vonk et al., 2014) producing additional $CO_2$.

### 3.3.4 The East Siberian Sea

The surface water $pCO_2$ was in equilibrium with the atmosphere or slightly higher in the well-stratified waters of the East Siberian Sea. Supersaturation was observed not only in the shallow shelf sea, as previously described in literature (Anderson et al., 2009, 2011; Pipko et al., 2005, 2011b; Semiletov et al., 2007, 2012), but also in the adjacent deep-water area. The likely causes of the detected $pCO_2$ distribution are the anomalous dynamics of atmospheric processes, in particular the deep low-pressure area over land and high-pressure area over the ocean, as well as the sharp reduction in sea ice coverage. This
led to RW being transported far to the north and northeast of the Eastern Arctic shelf and to intensive warming of the surface layer.

Surface water $pCO_2$ was somewhat higher at the oceanographic transect near the Lomonosov Ridge (the New Siberian Islands slope) than at the East Siberian Sea slope transect further to the east (Fig. 4); the $pCO_2$ of the East Siberian Sea slope transect was determined by different FW sources than was the $pCO_2$ of the Lomonosov Ridge. For the western transect river
runoff dominated as FW source, while sea ice MW contributed a considerable volume to the eastern transect (Fig. 7). In addition, the use of a four-component mixing model reveals the possible presence of significant concentrations (up to 25 % or more) of Pacific-derived waters at the eastern transect (Abrahamsen et al., 2009; Bauch et al., 2011). The possibility of Pacific-derived waters penetrating to the eastern slope of the Lomonosov Ridge has been discussed before (Makhotin, 2010 and references therein). The climatological circulation during 1997-2006 that was reconstructed using the 4D variational
approach shows the trajectories of several groups of Lagrangian particles (Shakhova et al., 2015); passive particles (or elements of the surface water mass) which originated in the eastern part of the East Siberian Sea between 160° and 170°E (the area impacted by the Pacific-derived waters) took less than two years to be transported to the Lomonosov Ridge area.

### 3.4 Dynamics of air-sea $CO_2$ fluxes in 2007



The air-sea $CO_2$ flux was computed with high spatial resolution along the cruise track from the Barents Sea to the East Siberian Sea (Fig. 9). Most of the studied areas of the ice-free waters served as a sink for atmospheric $CO_2$. Regions of intensive terrestrial impact in the Laptev Sea and the East Siberian Sea were the exception, and they acted as a weak source to the atmosphere (Fig. 9).

The Barents Sea is the strongest $CO_2$ sink in the Arctic region, yet estimates of the air-sea $CO_2$ flux in this area show large variability, depending on sub-region, season, and type of data used in the calculations (Lauvset et al., 2013). Using *daily* averaged wind and cubic gas-transfer velocity parametrization (Wanninkhof and McGillis, 1999) we estimated the air-sea $CO_2$ flux in the least-explored sub-region of the sea. The highest daily rates of $CO_2$ uptake were found in the northeastern Barents Sea (110 mmol $m^{-2}$ $day^{-1}$), where the high $\Delta pCO_2$ (around -150 µatm) coincided with high daily wind speed (15 m $sec^{-1}$). It should be noted that air-sea $CO_2$ fluxes were low, on the order of 9 mmol $m^{-2}$ $day^{-1}$, in the region between 25 and 45º E where the difference of $pCO_2$ values between sea and air was maximal (more than 150 µatm, Fig. 9), a result of the very low wind speed experienced here. Hence, it is obvious that it is the wind speed that yields the large spatial flux patchiness, because this parameter is much more variable in both time and space than is $\Delta pCO_2$.

Using the *hourly* wind speed and cubic gas-transfer velocity parametrization to compute the daily $CO_2$ flux (Wanninkhof and McGillis, 1999), a second region of maximum uptake became visible in the northwestern Kara Sea, despite the fact that in this region $\Delta pCO_2$ was almost two times lower (Fig. 9). Furthermore, the average $CO_2$ uptake rate was significantly greater in the Kara Sea than in the Barents Sea (50.5 versus 28.5 mmol $m^{-2}$ $day^{-1}$), using calculations based on the hourly wind speed, despite the fact that average $\Delta pCO_2$ was about half as much in the Kara Sea as it was in the Barents Sea (-46 µatm versus -95 µatm). The reason for this discrepancy is the wind speed, which averaged 9 m $sec^{-1}$ in the Barents Sea and 14 m $sec^{-1}$ in the Kara Sea (Fig. 9). Thus, the use of hourly wind speed to compute $CO_2$ fluxes takes into account small-scale parameter variations and improves the estimate of seawater $CO_2$ uptake capacity. In the eastern part of the Kara Sea, the ocean uptake was lower when the surface water $pCO_2$ was higher due to river discharge influence, which coincided with the weakening of the wind (Fig. 9).

Daily wind speed and quadratic parameterization of gas transfer velocity (Wanninkhof, 1992) were also used for calculating $CO_2$ fluxes in the northern Barents Sea. The $CO_2$ uptake intensity during the 2007 fall season reached an average of 113 g C $m^{-2}$ $year^{-1}$ and varied from 2 to 449 g C $m^{-2}$ $year^{-1}$. Determined by low water temperature and high wind speed, the obtained values were close to the maximum average $CO_2$ uptake in the southern and central Barents Sea in highly productive spring months (April and May) (Lauvset et al., 2013).

Lauvset et al. (2013) carefully assessed the seasonal cycle of air-sea $CO_2$ fluxes, but they did not cover the north of the sea comprehensively. Thus, the data obtained during our cruise adds information about the northern part of the sea, enabling a more accurate estimation of the absorption capacity of the whole Barents Sea in the fall season.

As noted before, there are both $CO_2$ sink and source regions in the Laptev and the East Siberian seas (Fig. 9). The southern Laptev Sea and the northwestern East Siberian Sea, where the terrestrial influence was significant and surface layer





temperatures were the highest, served as a weak source of $CO_2$ to the atmosphere. However, even if there were regions of large $\Delta pCO_2$ the fluxes were quite small (Fig. 9), a consequence of low winds; as a result the investigated area of the Laptev Sea as a whole was a weak sink for atmospheric $CO_2$ while that of the East Siberian Sea was a weak source of $CO_2$ to the

atmosphere (Table).

### 3.5 The interannual variability of pCO₂ and air-sea CO₂ fluxes

#### 3.5.1 The Barents and Kara seas

The surface layer of the Barents Sea was permanently undersaturated with respect to $CO_2$ and the ice-free waters were a sink of atmospheric $CO_2$, although the $CO_2$ flux was limited in the presence of sea ice cover (Figs. 3, 5, 9, Table). The lowest

surface water $pCO_2$ was observed in 2009 under temperature conditions close to freezing; the mean value (170 µatm) was less than half that of the atmosphere. The mean $pCO_2$ value was higher in 2009 (189 µatm) with maximal magnitude in 2007 (280 µatm). Nevertheless, the $CO_2$ uptake was higher in 2006 and even higher in 2007 (Table). This is an effect of less sea ice cover during investigations in 2006 relative to 2009 (sea ice concentration in the study area was ~50 and 80 %, respectively) and lack of sea ice cover in 2007. The higher wind speed in 2007 was an additional driver which increased $CO_2$

uptake.

The Kara Sea was also a sink of $CO_2$ in 2006, 2007, and 2009, but to a variable degree, from close to zero in sea-ice-covered or RW-influenced areas to -168 mmol m$^{-2}$ day$^{-1}$ in western ice-free waters. The surface water $pCO_2$ on the two stations carried out in 2006 in the eastern part of the Kara Sea was slightly above 250 µatm (mean value = 259 µatm). Note that RW was not found in the eastern Kara Sea in 2006 (Fig. 7). That was determined by cyclonic atmospheric circulation (Fig. 2),

which prevented spreading of the Ob and Yenisei river waters far to the east; sea ice MW was the main source of the FW in this region. The higher $pCO_2$ in 2007, increased from 284 to 372 µatm, was associated with higher sea surface temperature and the presence of the RW in the eastern part of the sea (Figs. 4, 7). Minimal surface $pCO_2$ (mean value 206 µatm) was revealed in 2009 (Fig. 5). Nevertheless, the average $CO_2$ uptake rate was the highest in 2007 (Table). Once again this is a result of the fact that the main part of the study area in 2009 was in sea-ice-covered waters while most of the 2007 cruise was

in open water (Fig. 1); wind was also stronger in 2007.

This finding stresses the importance of declining sea ice coverage and strengthening wind for the ocean's ability to take up atmospheric $CO_2$ in northern parts of the Barents and Kara seas, which are mainly remote from direct terrestrial discharge.

#### 3.5.2 The Laptev and East Siberian seas

The most extensive study over the three years was conducted in the Laptev Sea and in the northwestern East Siberian Sea,

the region where negative September ice concentration anomalies were most pronounced for the whole Arctic Ocean (ftp://sidads.colorado.edu). Surface waters of the outer shelf and slope in 2006 and 2009 were undersaturated in $CO_2$ relative to the atmosphere; 2007 was an exception (Table).





For comparative evaluation, we selected a transect north of the New Siberian Islands over the Lomonosov Ridge (Figs. 1, 10).

Of the three years, the observed surface water temperature was the highest and the salinity was the lowest in 2007 (Fig. 10). The low salinity was mainly related to the effective transfer of RW into the deep ocean due to the wind field and ice-free conditions, but salinity was also somewhat impacted by higher river discharge in 2007 (an average of 752, 822, and 738 km$^3$ for the Lena and Kolyma rivers in 2006, 2007, and 2009, respectively). The calculated RW content in the surface layer reached over 30 % in 2007, but did not exceed 20 % in 2006 or 10 % in 2009 (Fig. 11). Distribution of normalized $A_T$ also

confirmed the presence of a large amount of RW in 2007 (Fig. 11). The content of brine (indicating negative sea ice MW) in the surface water was also significantly higher during the summer season of 2007 compared to the fall seasons of 2006 and 2009 (Figs. 7, 11). This part of the Laptev Sea is known as a large sea ice production region and thus the brine signature builds up during the winter season. One consequence is that little sea ice MW was observed, up to a maximum of only 5 %, even though sea ice melt decreases the brine signal in the summer. The maximum MW fraction was in the low salinity range

of the surface water at the southern end of the section in 2006 (Figs. 7, 10, 11).

The substantial impact by river discharge in 2007 was characterized by high $pCO_2$, resulting in supersaturation along the transect. Thus, $CO_2$ out-gassing into the atmosphere was observed even on the northern edge of the outer shelf (Fig. 10). This situation is not typical for the deep waters of the Laptev Sea because the outer shelf normally acts as a sink for atmospheric $CO_2$, unlike the middle and inner shelves (Semiletov et al., 2007; Anderson et al., 2009; Pipko et al., 2016).

The $pCO_2$ in the lowest-salinity surface water was around 270 and 300 μatm in 2006 and 2009, respectively, although the runoff source ranged between 10 and 20 %. The higher $CO_2$ content of river runoff in 2007 not only contributed to high $pCO_2$ but also enhanced surface water temperature due to its high concentration of CDOM, which adsorbs solar radiation (Pugach et al., 2015; Semiletov et al., 2013). Except for the lower runoff content compared to 2007 the main contribution to the $pCO_2$ interannual variability was seawater temperature that was about 4 ℃ higher in 2007 (Fig. 10).

In summary, the strength and direction of air-sea $CO_2$ fluxes on the outer shelf and continental slope of the Laptev and the East Siberian seas varied significantly among years (Figs. 4, 5, 9, Table). The area was a sink of atmospheric $CO_2$ in 2006 and 2009, and the surface water of the East Siberian Sea was a weak source in 2007 (Table). That was related to the distribution of the RW plume on the shelf, the transport of terrestrial OM (and its oxidation to $CO_2$) by this plume, and the plume's impact on water temperature, as well as sea ice extent and wind speed (Fig. 1, Table). It should be noted that $pCO_2$

was higher in the deep regions of the Laptev and East Siberian seas than in the deep regions of the Kara Sea, even if the discharge is higher in the Kara Sea than in the ESAS. This is likely a combination of the dominating flow of the river plume and the source $pCO_2$ in the water that mixes with the runoff; the water in the Kara Sea comes from the Barents Sea with its low $pCO_2$, while the Laptev Sea is dominated by inflow from the Kara Sea.

**4 Conclusions**



This three-year study of the outer shelf and the continental slope waters of the Eurasian Arctic seas has revealed a general trend in the surface $pCO_2$ distribution, which manifested as an increase in $pCO_2$ values eastward, from the surface waters of the highly productive Barents Sea to the eastern Laptev Sea/western East Siberian Sea which are strongly influenced by terrestrial runoff. It has been shown that the influence of terrestrial discharge on the carbonate system of East Siberian Arctic sea surface waters is not limited to the shallow shelf. Furthermore, during certain meteorological conditions, the surface

waters of the outer shelf, as well as those of the continental slope of the East Siberian Arctic seas, can become supersaturated with respect to atmospheric $CO_2$.

The contemporary climate change affects air temperature in the Arctic region leading to sea-ice reduction and permafrost thawing, both on-land and off-shore. Growing air temperatures cause increased water temperature and strengthened wind activity, which intensifies water mass dynamics and air-sea exchange. It was shown that these changes in the Arctic

climatology can affect the capacity of this region to serve as a source or a sink for atmospheric $CO_2$ in two opposite ways: on the one hand, larger areas of open water due to sea ice reduction and longer ice-free periods can cause the outer shelf and slope of the West Siberian Eurasian Arctic seas (Barents and Kara seas) to develop a growing capability to absorb atmospheric $CO_2$; on the other hand, growing river discharge and degradation of permafrost, associated with thermal erosion of coasts and river banks, can increase the effectiveness of the East Siberian Arctic seas to act as a $CO_2$ source due to

increased terrestrial export of labile eroded carbon, a significant portion of which oxidizes to $CO_2$ (Semiletov et al., 2013, 2016). This was the situation in the deep regions of the Laptev and the East Siberian seas in 2007, when sea ice decline was especially pronounced, resulting in an increase of the $CO_2$ out-gassing area and a reduction of $CO_2$ absorption in the East Siberian Arctic seas.

This study has shown that contemporary climate change impacts the carbon cycle of the Eurasian Arctic Ocean and

influences air-sea $CO_2$ flux. It also highlighted the importance of considering small-scale variations in meteorological and hydrological parameters, varying both in time and in space, for estimating the air-sea exchange of $CO_2$. During these times of rapid environmental changes, results of this study stress the need for comprehensive multi-year investigations of dynamic deep-sea regions in order to estimate current and predict future capacity of the Arctic basin as a sink for atmospheric $CO_2$ based on high-resolution spatial coverage of the Arctic Ocean.

**Acknowledgements**

This work was supported by the Russian Government (grant 14.Z50.31.0012); the Far Eastern Branch of the Russian Academy of Sciences (FEBRAS); the International Arctic Research Center (IARC) of the University of Alaska Fairbanks through NOAA Cooperative Agreement NA17RJ1224; the U.S. National Science Foundation (Nos. OPP-0327664, OPP-0230455, ARC-1023281, ARC-0909546); and the NOAA OAR Climate Program Office (NA08OAR4600758). I.P. and S.P.

acknowledge the Russian Foundation for Basic Research (No. 14-05-00433a). L.G.A. and O.G. thank the Swedish Research Council and the Knut and Alice Wallenberg Foundation, as well as the European Research Council (ERC-AdG CC-TOP project #695331 to Ö.G.). N.S., O.D., I.P., and A.Ch. thank the Russian Science Foundation (grant No. 15-17- 20032). I.P., S.P., and I.S. would like to thank Dr. P.Ya. Tishchenko for collaboration and useful comments. We thank Candace O'Connor for English editing.





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

| | $\Delta pCO_2$, µatm | | | $U_{hourly}$, m sec$^{-1}$ | | | $F_{CO2}$,** mmol m$^{-2}$ day$^{-1}$ | | |
|---|---|---|---|---|---|---|---|---|---|
| | 2006 | 2007 | 2009 | 2006 | 2007 | 2009 | 2006 | 2007 | 2009 |
| Barents Sea | **-184 ± 25** (-211 ÷ -146) *n = 9* | **-95 ± 30** (-152 ÷ -10) *n = 172* | **-215 ± 31** (-246 ÷ -156) *n = 13* | **3.8 ± 1.8** (0.8 - 7.8) | **9.0 ± 3.8** (1.4 - 17.6) | **4.2 ± 2.1** (2 - 9) | **-3.2 ± 6.1** (-19.3 ÷ 0.0) | **-28.5 ± 33.4** (-160.8 ÷ -0.1) | **-0.6 ± 0.4** (-1.1 ÷ 0.0) |
| Kara Sea | **-114 ± 13.8** (-123 ÷ -104) *n = 2* | **-46 ± 21** (-91 ÷ -3) *n = 152* | **-179 ± 36** (-221 ÷ -109) *n = 19* | **9.9 ± 4.2** (6.9 - 12.8) | **14.3 ± 4.6** (1.5 - 21.7) | **8.4 ± 2.6** (2.5 - 13.6) | **-32.8 ± 31.8** (-55.3 ÷ -10.3) | **-50.5 ± 42.9** (-167.7 ÷ 0.0) | **-18.3 ± 19.4** (-71.0 ÷ 0.0) |
| Laptev Sea | **-95 ± 50** (-182 ÷ -2) *n = 15* | **-23 ± 40** (-118 ÷ 82) *n = 279* | **-101 ± 40** (-145 ÷ -42) *n = 16* | **4.7 ± 2.4** (0.4 - 11.0) | **5.0 ± 2.3** (0.6 - 12.1) | **6.2 ± 4.6** (1.0 - 16.2) | **-4.2 ± 6.4** (-17.8 ÷ 0.0) | **-0.9 ± 2.2** (-16.1 ÷ 6.5) | **-15.7 ± 34.5** (-136.2 ÷ 0.0) |
| East Siberian Sea | **-107 ± 12** (-117 ÷ -93) *n = 3* | **-4 ± 11** (-30 ÷ 18) *n = 295* | **-89 ± 6** (-101 ÷ -83) *n = 6* | **7.5 ± 3.9** (3.3 - 11.1) | **7.6 ± 3.3** (0.4 - 13.1) | **1.6 ± 0.8** (0.9 - 2.9) | **-5.8 ± 5.5** (-11.6 ÷ 0.8) | **0.2 ± 2.2** (-10.2 ÷ 8.4) | **-0.03 ± 0.0** (-0.1 ÷ 0.0) |

*Maximum variability of each parameter is shown in parentheses; the number of measurements (n) is shown in italics.
**Flux calculated according to Wanninkhof and McGillis (1999); negative values correspond to $CO_2$ flux into the ocean.





**(a)**

**(b)**

**Figure 1.** Ship routes and positions of oceanographic station in the study area. (**a**) Positions of oceanographic stations performed in 2006, 2007, and 2009 are marked as colored circles: 2006 - blue; 2007 – red; 2009 – green. Position of the sea-ice edge during the expeditions is marked as colored curved lines: 2006 - blue; 2007 - red; 2009 - green; (**b**) the ship's route, along which the high-frequency measurements were performed in 2007.



**(a)**                                        **(b)**

**(c)**

**Figure 2.** Sea level pressure fields (mbar) averaged over the summer season of 2006 (**a**), 2007 (**b**), and 2009 (**c**) from NCEP data (www. esrl.noaa.gov).





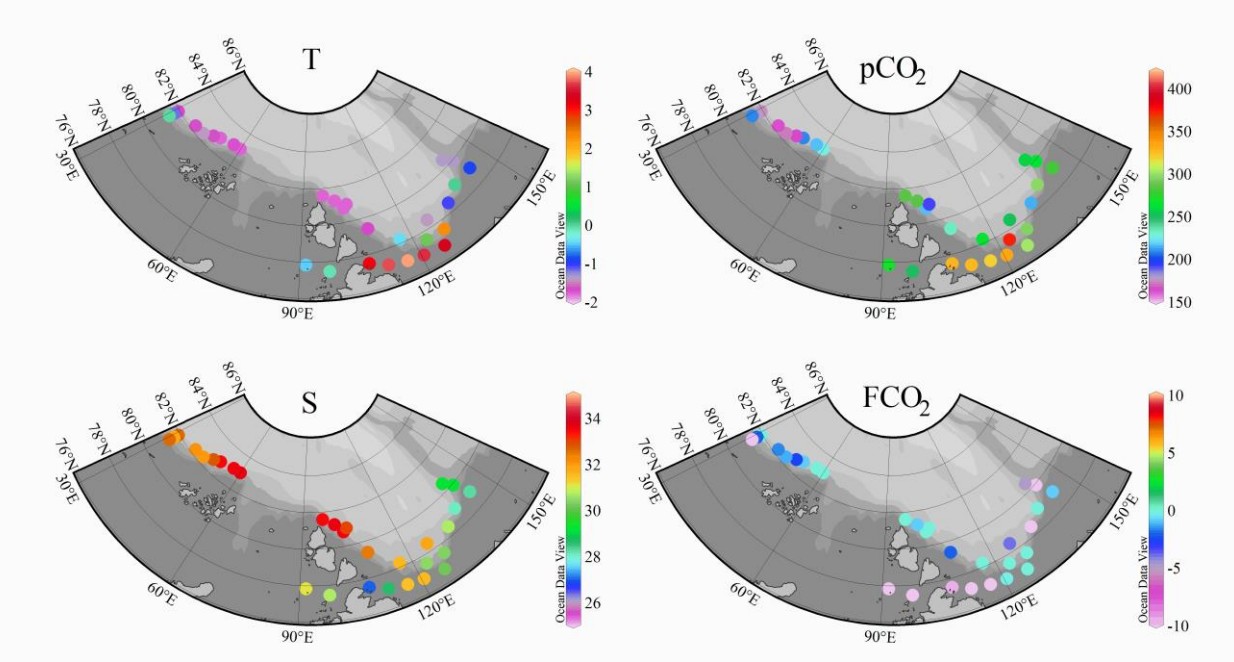

**Figure 3.** Spatial distribution of sea surface temperature (T, ºC), salinity (S), pCO$_2$ (µatm), and air-sea CO$_2$ fluxes (F$_{CO2}$, mmol m$^{-2}$ day$^{-1}$) during the 2006 study.



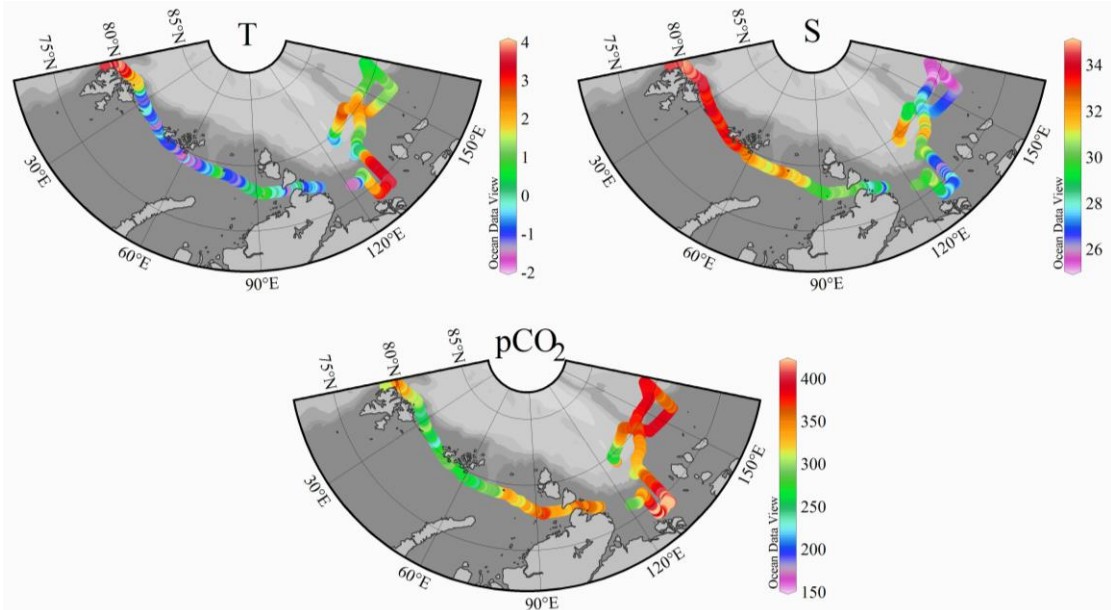

**Figure 4.** Spatial distribution of sea surface temperature (T, °C), salinity (S) and pCO$_2$ (µatm) during the 2007 study.



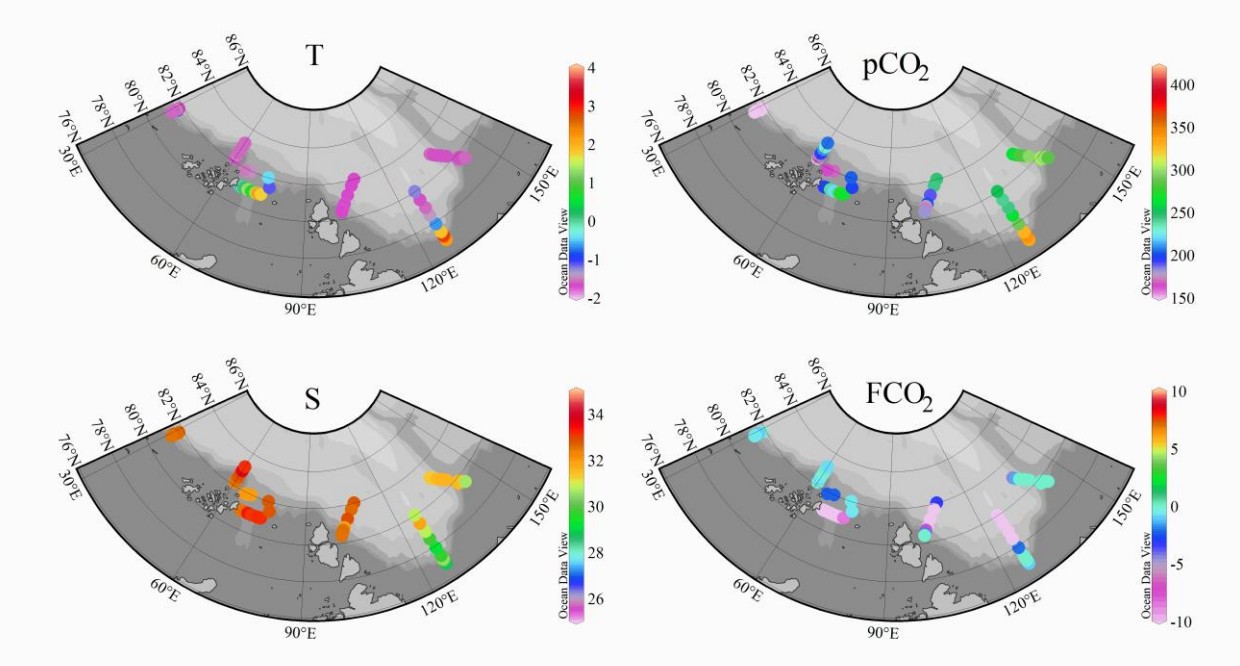

**Figure 5.** Spatial distribution of sea surface temperature (T, ºC), salinity (S), $pCO_2$ (µatm), and air-sea $CO_2$ fluxes ($F_{CO2}$, mmol m$^{-2}$ day$^{-1}$) during the 2009 study.





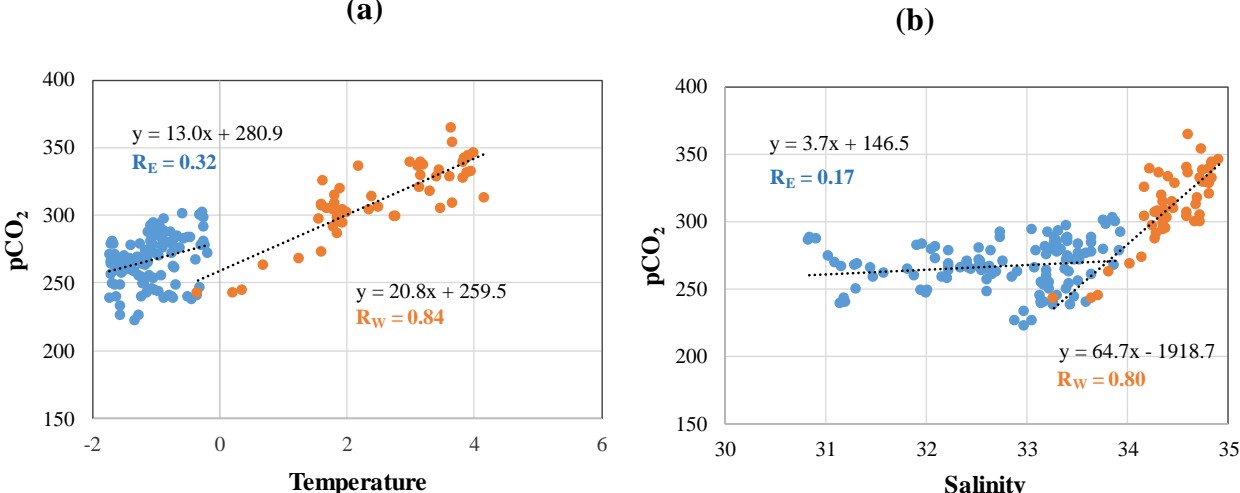

**Figure 6.** Relationship between $pCO_2$ (µatm) and temperature (°C) (**a**), and salinity (**b**) in the northern Barents Sea: western part – W, orange color, eastern part – E, blue color.





**Figure 7.** Spatial distribution of fractions of river water (RW, %) and sea-ice meltwater (MW, %) in surface water during the 2006 (**a**), 2007 (**b**), and 2009 (**c**) studies.





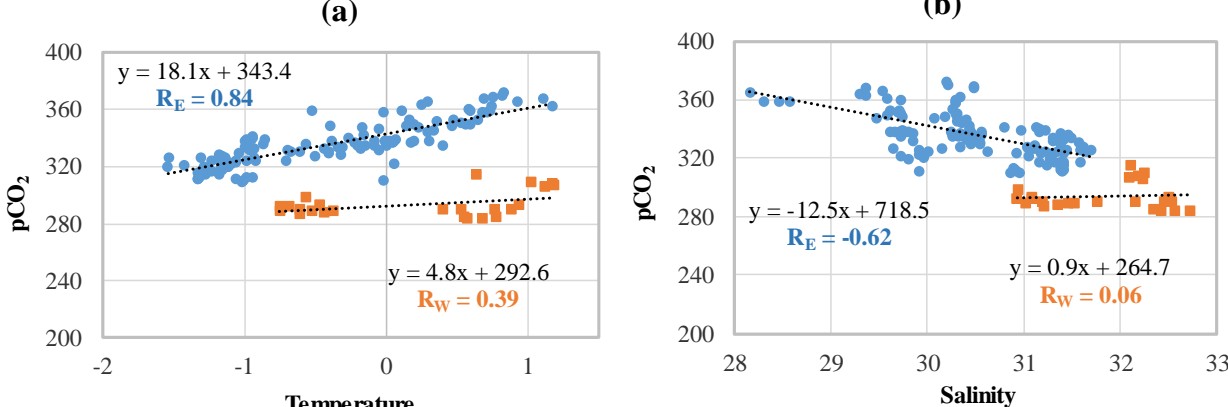

**Figure 8.** Relationship between pCO$_2$ (µatm) and temperature (°C) (**a**) and salinity (**b**) for samples collected in the western (W, orange color) and eastern (E, blue color) Kara Sea during the 2007 study.



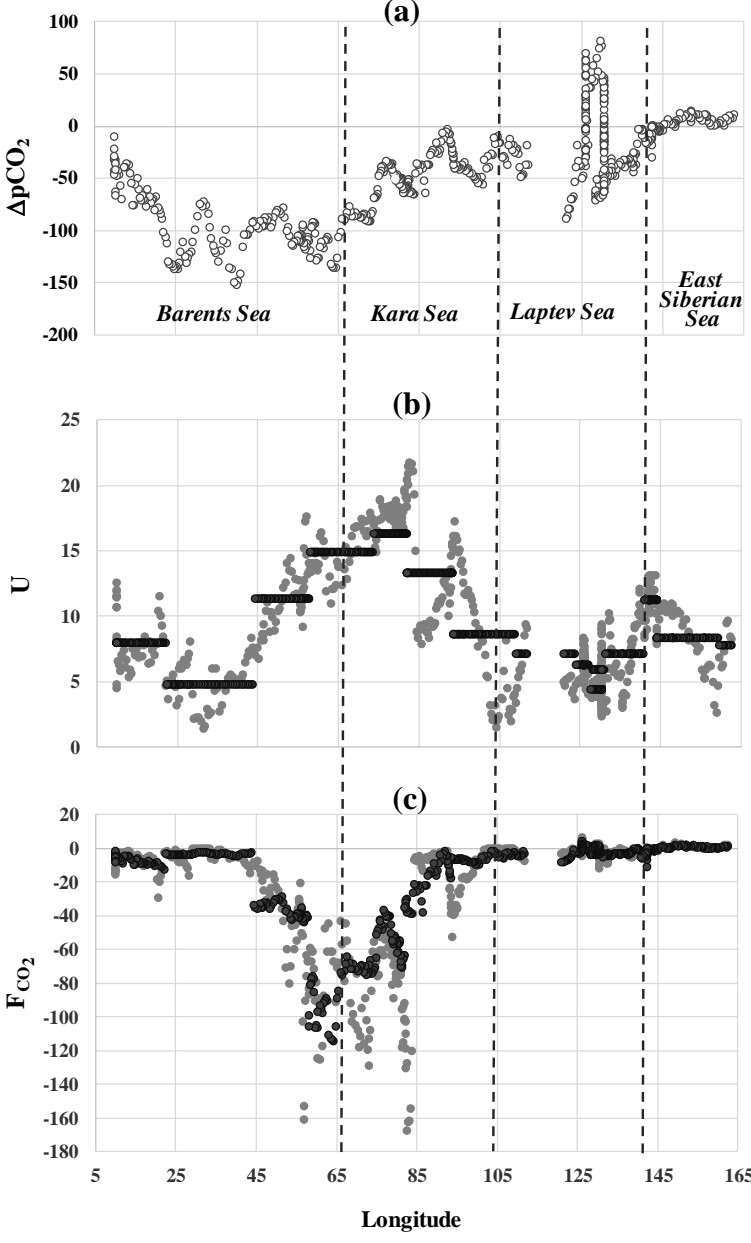

**Figure 9.** Distribution of (**a**) $\Delta pCO_2$ (µatm), (**b**) wind speed (U, m s$^{-1}$), and (**c**) air-sea $CO_2$ fluxes ($F_{CO2}$, mmol m$^{-2}$ day$^{-1}$) along the ship's route in 2007; grey color corresponds to the hourly averaged wind speed in panel (**b**) and the hourly-based air-sea $CO_2$ fluxes in panel (**c**); black color corresponds to the daily averaged wind speed in panel (**b**) and the daily average-based air-sea $CO_2$ fluxes in panel (**c**).





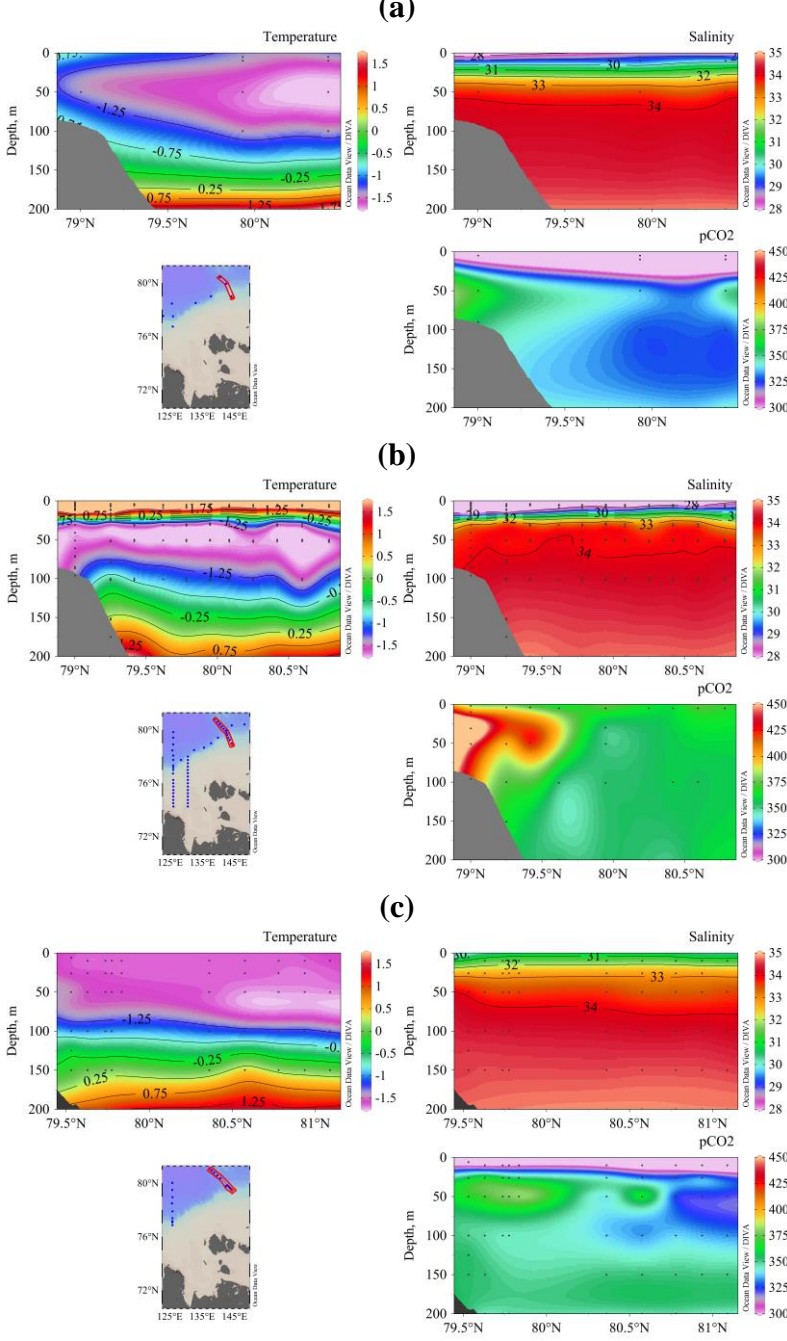

**Figure 10.** Distribution of temperature (°C), salinity, and pCO₂ (µatm) along the transect across the continental slope of the New Siberian Islands at ~140-145°E during the 2006 (**a**), 2007 (**b**), and 2009 (**c**) studies.





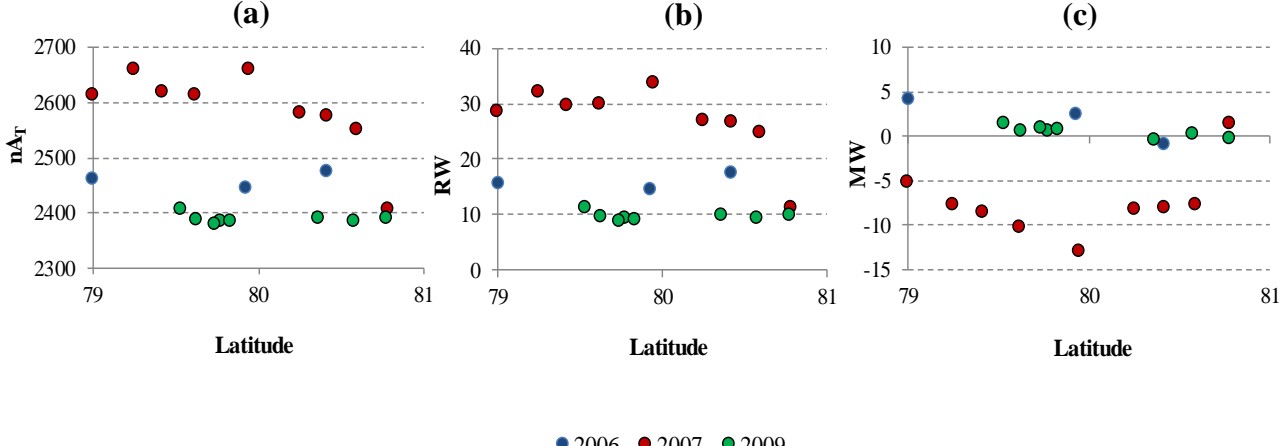

**Figure 11.** Distribution of (**a**) surface normalized total alkalinity (nA$_T$, μmol kg$^{-1}$) and fractions (**b**) of river water (RW, %) and (**c**) sea-ice meltwater (MW, %) along the transect across the continental margin of the New Siberian Islands at ~140-145$^o$E in 2006, 2007, and 2009.