# Peer review of "The spatial and inter-annual dynamics of the surface water carbonate system and air-sea CO2 fluxes in the outer shelf and slope of the Eurasian Arctic Ocean"

_Ocean Science, 2017_

## Referee Comment (RC1) · Anonymous Referee #1 · 8 Jun 2017

This paper presents pCO2 data and associated air-sea flux of CO2 from the Eurasian sector of the Arctic Ocean for three years (2006, 2007 and 2009). Data in this region are extremely scarce due to the logistical difficulties involved. As such, this paper makes a valuable contribution to our understanding of CO2 exchange between the atmosphere and the Arctic Ocean at a time when the latter is undergoing rapid change. The authors have followed up various lines of thought to explain inter-annual and regional differences. I particularly liked the separation and apportionment of freshwater sources (MW and RW). The work presented is substantial, the analysis is very thorough and the paper is well structured and well written. The written style varies slightly between sections, probably reflecting the fact that different authors had written different

section – a consistent writing style may slightly improve the manuscript in this respect. I enjoyed this paper and have no hesitation to recommend its publication. I have some specific comments which are outlined below. I would leave most of my comments at the authors' discretion, but I would urge them to address comments 6 to 9 in particular. Specific Comments:

1) Lines 79-84: A couple of useful references might also be: Mann et al., 2012, doi:10.1029/2011JG001798 and Mann et al., 2015, doi: 10.1038/ncomms8856

2) Line 129: At what depth was the intake for the pumped seawater?

3) Line 135: Please give batch numbers for carbonate CRMs

4) Line 146-147: Does the 30-minute averaging have an effect on accuracy? Over 30 minutes a moving ship may cross fronts, river-plumes, marginal ice zones etc. Averaging would therefore smooth if not obscure any gradients in pCO2.

5) Line 255: The high Oxygen supersaturation observed in the Barents Sea is intriguing. Clearly, temperature alone explains 84% of the variance in pCO2 and the authors are correct to point out the air-sea exchange may not have fully compensated for earlier biological drawdown of CO2. Typically, the turnover of the surface mixed layer CO2 via gas-exchange is in the order of months because of carbonate buffering. In contrast, Oxygen will re-equilibrate with the atmosphere in days/weeks. Simultaneous CO2 undersaturation and O2 oversaturation would therefore suggest very recent PP. Satellite Chlorophyll might give additional insight should the authors wish to expand their analysis.

6) Line 318-320: The authors state that "optically-active OM and suspended material... promotes the accumulation of solar radiation... which decreases the heat content [leading to further ice melt]". I don't think that OM and SPM contribute hugely to the heat content of surface waters. The big switch from high albedo with ice-cover to low albedo in ice-free water would have a much bigger effect than the absorbing constituents such

as OM. Nevertheless, I do believe that OM is hugely relevant here since OM will undergo photolysis to CO2 (e.g. Mann et al., 2012, doi:10.1029/2011JG001798). This is in addition to microbial OM mineralization which the authors have covered.

7) Line 330: In relation to pCO2 supersaturation in the East Siberian Sea, the authors state that this was due to the atmospheric pressure gradient which diverted river water offshore. I presume that this would carry high OM to the ESS which would be further mineralized to CO2, hence the elevated pCO2. It would be worth stating this explicitly as the current text leaves it up to the reader to make that connection. If the reader fails to make the connection, then the message is lost.

8) Line 351-355: Regarding the flux of CO2, the authors state that the highest influx coincided with high wind, while the highest DpCO2 did not result in very high influx because of low wind. The authors have used the cubic relationship of Wanninkhoff and McGillis (1999) between k and wind speed for calculating the flux. There is a wide spectrum of Kw-wind relationships and the one used here returns k values at the upper end of the range. I wonder whether a middle of the range formulation might be better while the separate debate regarding the Kw-wind relationship goes on in the air-sea gas exchange community. Wanninkhoff, 1992 would certainly be ok here. On line 190 it is stated that the W92 formulation was used, but there is no further mention of it in the results (?).

9) Lines 363-364: The authors state that hourly wind speed improves the estimate of CO2 uptake capacity (line 363-364). I have a technical objection to the use of the term "uptake capacity" (or "uptake intensity" on line 368). What do these mean? Surely, we are talking about "flux", so why not stick to that term and avoid ambiguity? Whether hourly wind-speed improves the estimate of the flux is somewhat irrelevant given that the flux depends so much on one's choice of k formulation. This alone makes a difference of 50%, if not more at high wind speeds. Each parameterization of Kw has its limitations and is calculated over different time-scales so it may or may not be appropriate to apply this to hourly wind data. I would simplify this discussion by not going

into such details of air-sea exchange. In my opinion, it's fine to clearly state how the flux is calculated here and move on to the other sections. Statements regarding improvements of the flux by hourly vs. daily wind speed are beyond the scope of this paper.

10) Figure 1: It would be informative to also plot the 2007 sea-ice extent on panel b of Figure 1.

---

## Referee Comment (RC2) · Anonymous Referee #2 · 21 Jul 2017

Interactive comment on "The dynamics of carbon dioxide system in the outer shelf and slope of the Eurasian Arctic Ocean" by I. Pipko et al.

The paper illustrates the surface pCO2 distributions in the Arctic Ocean and the associated air sea CO2 fluxes within wide and shallow shelves of the Eurasian sector, whih can be affected by intense exchanges at the air sea interface. In addition, spatial and temporal variabilities are presented together with different drivers of the marine carbonate system in one of the most sensitive region to climate change and ocean acidification. The region has been undergoing rapid changes for the last decades. The collected data refers to three seasonal campaignes, conducted in late summer/fall

2006, 2007 and 2009, characterized by different meteorological conditions. The spatio-temporal variability and the different drivers are thoroughly analyzed and well discussed, while results are clearly presented. In my opinion the objectives of this study are clearly presented and fully reached. The paper can add valuable contribution to the knowlwdge of CO2 fluxes in a polar region, where dearth of data is limiting. I enjoyed the paper, in particular the introduction and the discussion on the response of marine carbonate system to the different drivers well enlighting the complexity of the system. I believe it is worth of publication. Nevertheless I would recommend some minor revisions summarized in the specific comments.

Specific comments:

1) It seems to me that title does not fully mirror the focus of the paper, mainly addressed to the upper layer properties, distributions and dynamics. . .. If you agree would you mind suggesting this even in the title?

2) Line 30: more caution should be used about "a growing CO2 evasion occurs . . .." as the estimated fluxes from the sea to the atmosphere (in Tab 1) are really very low ! Wanninkhof and McGillis (1999) are reported to underestimate fluxes at low wind speed, that seems the case. I don't mean to open discussion about the best parameterisation (for instance Nightingale et al. 2000 might be suggested). I accept the author's choice but please be cautious about results. I rather would say that uptake was strongly weakening under 2007 environmental conditions as surface seawater appears in equilibrium with atmosphere . . .

3) Paragraph 2.2.2: author should provide the temperature conditions of analysis. Titration has been performed at costant temperature ? and which one ? Due to the variety of analytical methods and measurement units, the international community working on marine carbonate system has decided to adopt common protocols (requiring the analysis at constant temperature, and common measurement units) Protocols reported by Dickson et al 2007 that authors cite, are recommended.

4) Paragraph 2.2.3: indicate the scale of the pH measurement and again the temperature. The international community working on CO2 fluxes, ocean acidification and impacts, has decided to adopt common protocols and common measurement units in order to increase utilization of data among different scientific communities. This unifomity would increase a wider utility of the paper. Protocols reported by Dickson et al 2007 that authors cite, are recommended. Please refer to them for units and scale. Line 140-141: authors shoul provide the accuracy of the method , for consistency with TA. This can be done by calibration against the reference materials (CRM's supplied by Dickson) and using CO2SYS for calculating the pHT of CRMs at the temperature of analysis.

Specific comments at point 3 and 4 are necessary also for the next paragraph (2.2.4), where CO2SYS programme is mentioned. This could be useful to non expert (of carbonate system analysis) readers.

5) Paragraph 2.2.4, lines 148-149: in order to prevent misundersting and not confuse direct continuous pCO2 measurements (by SAMI CO2 sensor) with the calulated pCO2 from discrete samples (collected by Rosette), I suggest to specify "At oceanographic stations surface pCO2 values were calculated, on discrete samples, from pHT25, AT and inorganic nutrients data using CO2SYS..." In addition authors should say which constants for sulfate and borate (KSO4 and KBorate) have been choosen in the CO2SYS programme.

6) Lines 367-374: rephrase the two paragraphs as "In order to compare our estimates with those calculated by Lauvset et al. (2013) which carefully assessed the seasonal cycle of air-sea CO2 fluxes in the Barents Sea, daily wind speed and quadratic parameterization of gas transfer velocity (Wanninkhof, 1992) were used for calculating CO2 fluxes in the northern Barents Sea. The CO2 uptake during the 2007 fall season reached an average ….. As the dataset by Lauvset et al. (2013) did not cover the north of the sea comprehensively, the data obtained during our cruise adds information enabling more accurate estimation of the absorption capacity of the whole Barents

Sea in the fall season."

7) Lines 408-409: again I feel necessity of a clear indication that pCO2 data of the selected transect, reported in fig 10, are calculated for discrete samples (from AT, pHT25 and inorganic nutrients data) by means of CO2SYS programme.

8) Fig 10 seems underutilezed in the text, as only surface data are compared without any further discussion about vertical distributions. As the figure is very informative could you please comment a bit more ?

9) Line 421: I find a bit "dangerous" using here the word "supersaturation" as this make me to wonder if supersaturation has been really computed (as

10) Line 422: I find not fully proper to say that CO2 outgassing into the atmosphere was observed …. (Fig 10), as the calculated fluxes for the Laptev and Est Siberian seas were really very low (see Tab 1). I would prefer rephrase as "Thus $\Delta$pCO2 conditions (Tab 1) favouring CO2 outgassing into the atmosphere were observed"

11) Line 456-458: I suggest authors to rephrase as "… resulting in an increase of the area where seawater pCO2 was in equilibrium with atmosphere and consequent reduction of CO2 adsorption in the East Siberian Arctic seas".

---

## Author Comment (AC1) · 14 Aug 2017

**Response to review comments on "The dynamics of the carbon dioxide system in the outer shelf and slope of the Eurasian Arctic Ocean" by Irina I. Pipko et al.**

**Anonymous Referee #1**

This paper presents $pCO_2$ data and associated air-sea flux of $CO_2$ from the Eurasian sector of the Arctic Ocean for three years (2006, 2007 and 2009). Data in this region are extremely scarce due to the logistical difficulties involved. As such, this paper makes a valuable contribution to our understanding of $CO_2$ exchange between the atmosphere and the Arctic Ocean at a time when the latter is undergoing rapid change. The authors have followed up various lines of thought to explain inter-annual and regional differences. I particularly liked the separation and apportionment of freshwater sources (MW and RW). The work presented is substantial, the analysis is very thorough and the paper is well structured and well written. The written style varies slightly between sections, probably reflecting the fact that different authors had written different section – a consistent writing style may slightly improve the manuscript in this respect. I enjoyed this paper and have no hesitation to recommend its publication. I have some specific comments, which are outlined below. I would leave most of my comments at the authors' discretion, but I would urge them to address comments 6 to 9 in particular.

We would like to thank Anonymous Referee #1 for his thoughtful and positive review as well as helpful advices to improve our manuscript. Our responses to all of the Referee's comments are shown in blue below.

**Specific Comments**:
1) Lines 79-84: A couple of useful references might also be: Mann et al., 2012, doi:10.1029/2011JG001798 and Mann et al., 2015, doi: 10.1038/ncomms8856.
Thanks, references will be added.

Mann, P. J., Davydova ,A., Zimov, N., Spencer, R. G. M., Davydov, S., Bulygina, E., Zimov, S., and Holmes, R. M. (2012). Controls on the composition and lability of dissolved organic matter in Siberia's Kolyma River basin, J. Geophys. Res., 117, G01028, doi:10.1029/2011JG001798.

Mann, P.J., Eglinton, T.I., McIntyre, C.P., Zimov, N., Davydova, A., Vonk, J.E., Holmes, R.M., and Spencer, R.G.M. (2015). Utilization of ancient permafrost carbon in headwaters of Arctic fluvial networks, Nature Communications, 6, 7856, doi: 10.1038/ncomms8856.

2) Line 129: At what depth was the intake for the pumped seawater?

Seawater was taken from a depth of about 4 m. This information will be added to the text.

3) Line 135: Please give batch numbers for carbonate CRMs
Batch #96 was used in 2009, in the 2006 and 2007 cruises the hydrochloric acid concentration was determined using a standard solution of $Na_2CO_3$ made up by carefully weighing $Na_2CO_3$ of 99.995% purity (DOE, 1994; Pavlova et al., 2008).
This information will be added in the manuscript.

4) Line 146-147: Does the 30-minute averaging have an effect on accuracy? Over 30 minutes a moving ship may cross fronts, river-plumes, marginal ice zones etc. Averaging would therefore smooth if not obscure any gradients in $pCO_2$.
Thank you for pointing this out. For comparison, plots of $pCO_2$ (and hydrological parameters) with 15 min averaging have been constructed (Figure 1). Comparison of the graphs did not reveal additional features in the distribution of these parameters.
Nevertheless, we will use the 15-min averaging for a more detail presentation of the available data.

[Figure]

Figure 1. Distribution of $pCO_2$ and salinity: 30-min averaged –upper panels, 15-min averaged – bottom panels.

5) Line 255: The high Oxygen supersaturation observed in the Barents Sea is intriguing. Clearly, temperature alone explains 84% of the variance in $pCO_2$ and the authors are correct to point out

the air-sea exchange may not have fully compensated for earlier biological drawdown of $CO_2$. Typically, the turnover of the surface mixed layer $CO_2$ via gas-exchange is in the order of months because of carbonate buffering. In contrast, Oxygen will re-equilibrate with the atmosphere in days/weeks. Simultaneous $CO_2$ undersaturation and $O_2$ oversaturation would therefore suggest very recent PP. Satellite Chlorophyll might give additional insight should the authors wish to expand their analysis.

Thank you for suggestion. We have noted the remaining effect from primary production late in the season, i.e. close to our study. Unfortunately, we do not have field information regarding the distribution of chlorophyll-a and oxygen concentrations in the Barents Sea in autumn 2007, and the available satellite images do not allow to reliably estimating the intensity of photosynthetic processes throughout the photic zone. To avoid confusion, we will remove an information about the values of oxygen saturation observed in 2006 and 2009.

6) Line 318-320: The authors state that "optically-active OM and suspended material... promotes the accumulation of solar radiation... which increases the heat content [leading to further ice melt]". I don't think that OM and SPM contribute hugely to the heat content of surface waters. The big switch from high albedo with ice-cover to low albedo in ice-free water would have a much bigger effect than the absorbing constituents such as OM.

Sure, the reduction in albedo due to increase in the area of ice-free water is a main driving factor in increasing the heat content of surface waters. However, the elevated concentrations of CDOM and SPM also can contribute to the surface waters heating and subsequent melting of sea ice by absorbing shortwave visible radiation (Granskog et al., 2007; Hill, 2008; Logvinova et al., 2016). For example, Granskog with co-authors (2015) noted that high concentrations of CDOM in the surface polar waters resulted in 50–60% more heat deposition in the upper meters relative to clearest natural waters.

We will add these references to support this statement:

Granskog, M. A., A. K. Pavlov, S. Sagan, P. Kowalczuk, A. Raczkowska, and C. A. Stedmon (2015). Effect of sea-ice melt on inherent optical properties and vertical distribution of solar radiant heating in Arctic surface waters, J. Geophys. Res. Oceans, 120, doi:10.1002/2015JC011087.

Granskog, M.A., Macdonald, R.W., Mundy, C.J., Barber, D.G. (2007). Distribution, characteristics and potential impacts of chromophoric dissolved organic matter (CDOM) in Hudson Strait and Hudson Bay, Canada. Cont. Shelf Res. 27, 2032–2050.

Hill, V.J. (2008). Impacts of chromophoric dissolved organic material on surface ocean heating in the Chukchi Sea. J. Geophys. Res. 113, C07024. http://dx.doi. org/10.1029/2007JC004119.

Logvinova, C. L., Frey, K.E. and Cooper, L.W. (2016). The potential role of sea ice melt in the distribution of chromophoric dissolved organic matter in the Chukchi and Beaufort Seas, Deep-Sea Research II, 130 28–42.

Nevertheless, I do believe that OM is hugely relevant here since OM will undergo photolysis to $CO_2$ (e.g. Mann et al., 2012, doi:10.1029/2011JG001798). This is in addition to microbial OM mineralization which the authors have covered.

Thank you for pointing this out. We did not pay an enough attention to the role of photolysis, because we assumed that biomineralization is the dominant mechanism for removal of terrestrial DOM (Belanger et al., 2006; Fichot and Benner, 2014; Kaiser et al., 2017), and some authors demonstrate that sunlight exposure does not substantially degrade DOM on Arctic shelves (~1% DOC loss, Osburn et al., 2009). On the contrary, photomineralisation has important role in the Siberian Rivers, particularly in samples collected during the spring freshet (Mann et al., 2012). We will note in the manuscript that photochemical transformation of terrestrial DOC and direct photomineralisation of OM also has an effect on increasing concentrations of $CO_2$ in surface waters.

Bélanger, S., H. Xie, N. Krotkov, P. Larouche, W. F. Vincent, and Babin, M. (2006). Photomineralization of terrigenous dissolved organic matter in Arctic coastal waters from 1979 to 2003: Interannual variability and implications of climate change, Global Biogeochem. Cycles, 20, GB4005, doi:10.1029/2006GB002708.

Fichot, C. G., and Benner, R. (2014). The fate of terrigenous dissolved organic carbon in a river-influenced ocean margin, Global Biogeochem. Cycles, 28, doi:10.1002/2013GB004670.

Kaiser, K., Benner, R., and Amon, R. M. W. (2017). The fate of terrigenous dissolved organic carbon on the Eurasian shelves and export to the North Atlantic, J. Geophys. Res. Oceans, 122, 4–22, doi:10.1002/2016JC012380.

Osburn, C. L., Retamal, L., and Vincent, W. F. (2009). Photoreactivity of chromophoric dissolved organic matter transported by the Mackenzie River to the Beaufort Sea, Marine Chemistry, 115, 10–20.

7) Line 330: In relation to $pCO_2$ supersaturation in the East Siberian Sea, the authors state that this was due to the atmospheric pressure gradient which diverted river water offshore. I presume that this would carry high OM to the ESS which would be further mineralized to $CO_2$, hence the elevated $pCO_2$. It would be worth stating this explicitly as the current text leaves it up to the reader to make that connection. If the reader fails to make the connection, then the message is lost.

We will clarify this point in the paper.

8) Line 351-355: Regarding the flux of $CO_2$, the authors state that the highest influx coincided with high wind, while the highest $DpCO_2$ did not result in very high influx because of low wind. The authors have used the cubic relationship of Wanninkhof and McGillis (1999) between k and wind speed for calculating the flux. There is a wide spectrum of Kw-wind relationships and the one used here returns k values at the upper end of the range. I wonder whether a middle of the range formulation might be better while the separate debate regarding the Kw-wind relationship goes on in the air-sea gas exchange community. Wanninkhof, 1992 would certainly be ok here.

We used a cubic relationship between gas exchange and wind speed (Wanninkhof and McGillis, 1999) because this parametrization is appropriate for short-term winds (as a quadratic dependence of gas exchange on wind speed (Wanninkhof, 1992)). Moreover, a cubic relationship (Wanninkhof and McGillis, 1999) demonstrates a better agreement with eddy covariance $CO_2$ flux measurements in comparison with Wanninkhof, 1992 parametrization over the East Siberian and Laptev seas in the late summer season (Pipko et al., 2008). We agree with Referee that each relationship has its limitations and uncertainties. However, even if we used W92 parametrization, we found that the highest $CO_2$ fluxes were not coincide with maximum in $\Delta pCO_2$ (Figure 2).

Pipko, I.I., Repina, I.A., Salyuk, A.N., Semiletov, I.P., and Pugach, S.P. (2008). Comparison of Calculated and Measured $CO_2$ Fluxes between the Ocean and Atmosphere in the Southwestern Part of the East Siberian Sea, Doklady Earth Sciences, 422, 7, 1105-1108.

[Figure]

Figure 2. Distribution of $\Delta pCO_2$ ($\mu atm$) (A), wind speed (U, m s$^{-1}$) (B), and air-sea $CO_2$ fluxes ($F_{CO2}$, mmol m$^{-2}$ day$^{-1}$) (C –for cubic parametrization (Wanninkhof and MacGillis, 1999), D – for quadratic parametrization (Wanninkhof, 1992)) along the ship's route in 2007. Grey color corresponds to the hourly averaged wind speed and the hourly-based air-sea $CO_2$ fluxes; black color corresponds to the daily averaged wind speed and the daily average based air-sea $CO_2$ fluxes.

On line 190 it is stated that the W92 formulation was used, but there is no further mention of it in the results (?).

We used $CO_2$ flux calculations based on Wanninkhof, 1992 formulation for comparison with Lauvset et al. (2013) data for autumn 2007 (Lines 367-371).

9) Lines 363-364: The authors state that hourly wind speed improves the estimate of $CO_2$ uptake capacity (line 363-364). I have a technical objection to the use of the term "uptake capacity" (or "uptake intensity" on line 368). What do these mean? Surely, we are talking about "flux", so why not stick to that term and avoid ambiguity?

Thank you, we will replace these terms.

Whether hourly wind-speed improves the estimate of the flux is somewhat irrelevant given that the flux depends so much on one's choice of k formulation. This alone makes a difference of 50%, if not more at high wind speeds. Each parameterization of Kw has its limitations and is calculated over different time-scales so it may or may not be appropriate to apply this to hourly wind data. I would simplify this discussion by not going into such details of air-sea exchange. In my opinion, it's fine to clearly state how the flux is calculated here and move on to the other sections. Statements regarding improvements of the flux by hourly vs. daily wind speed are beyond the scope of this paper.

We agree with the reviewer, and will remove this part of the discussion from the text.

10) Figure 1: It would be informative to also plot the 2007 sea-ice extent on panel b of Figure 1.

Thank you, it will be added.

---

## Author Comment (AC2)

**Response to review comment on "The dynamics of the carbon dioxide system in the outer shelf and slope of the Eurasian Arctic Ocean" by Irina I. Pipko et al.**

**Anonymous Referee #2**

The paper illustrates the surface $pCO_2$ distributions in the Arctic Ocean and the associated air sea $CO_2$ fluxes within wide and shallow shelves of the Eurasian sector, which can be affected by intense exchanges at the air sea interface. In addition, spatial and temporal variabilities are presented together with different drivers of the marine carbonate system in one of the most sensitive region to climate change and ocean acidification. The region has been undergoing rapid changes for the last decades. The collected data refers to three seasonal campaigns, conducted in late summer/fall 2006, 2007 and 2009, characterized by different meteorological conditions. The spatiotemporal variability and the different drivers are thoroughly analyzed and well discussed, while results are clearly presented. In my opinion, the objectives of this study are clearly presented and fully reached. The paper can add valuable contribution to the knowledge of CO2 fluxes in a polar region, where dearth of data is limiting. I enjoyed the paper, in particular the introduction and the discussion on the response of marine carbonate system to the different drivers well enlighting the complexity of the system. I believe it is worth of publication.

We would like to thank Anonymous Referee #2 for his thoughtful and positive review as well as helpful advices to improve our manuscript. Our responses to all of the Referee's comments are shown in blue below.

Specific comments:

1)      It seems to me that title does not fully mirror the focus of the paper, mainly addressed to the upper layer properties, distributions and dynamics. . .. If you agree would you mind suggesting this even in the title?

We followed the suggestion; the title has been changed for:

"The spatial and inter-annual dynamics of the surface waters carbon dioxide system and air-sea $CO_2$ fluxes in the outer shelf and slope of the Eurasian Arctic Ocean".

2)      Line 30: more caution should be used about "a growing $CO_2$ evasion occurs . . .." as the estimated fluxes from the sea to the atmosphere (in Tab 1) are really very low ! Wanninkhof and McGillis (1999) are reported to underestimate fluxes at low wind speed, that seems the case. I don't mean to open discussion about the best parameterisation (for instance Nightingale et al. 2000 might be suggested). I accept the author's choice but please be cautious about results. I

rather would say that uptake was strongly weakening under 2007 environmental conditions as surface seawater appears in equilibrium with atmosphere . . .

The text will be re-written as:

"In contrast, the uptake of $CO_2$ was strongly weakening in the outer shelf and slope waters of the East Siberian Arctic seas during the 2007 environmental conditions. The surface seawater appears in equilibrium or slightly supersaturated by $CO_2$ relative to atmosphere because of increasing influence of river runoff and its input of terrestrial organic matter that mineralizes, in combination with the high surface-water temperature during sea ice-free conditions."

3) Paragraph 2.2.2: author should provide the temperature conditions of analysis. Titration has been performed at constant temperature ? and which one ? Due to the variety of analytical methods and measurement units, the international community working on marine carbonate system has decided to adopt common protocols (requiring the analysis at constant temperature, and common measurement units) Protocols reported by Dickson et al 2007 that authors cite, are recommended.

Thank you for pointing this out. The text will be re-written as:

"Samples for $A_T$ were analyzed in the lab within one month using an indicator titration method in which 25 ml of seawater was titrated with 0.02 M HCl in an open cell according to (Bruevich, 1944; Pavlova et al., 2008). Measurements were performed at 20°C, with the temperature in the cell controlled to within 0.1°C. In 2000 the Carbon Dioxide in the Ocean working group of the North Pacific Marine Science Organization (PICES) performed an intercalibration of $A_T$ in seawater using CRMs. The results of the intercalibration showed that the alkalinity values obtained by the Bruevich method are in agreement with the standard within ±1 µmol kg$^{-1}$ when state-of-the-art analytical practice is applied (Pavlova et al., 2008)".

Pavlova, G. Yu., Tishchenko, P. Ya., Volkova, T. I., Dickson, A., and Wallmann, K. (2008) Intercalibration of Bruevich's method to determine the total alkalinity in seawater, Oceanology, 48, 3, 438-443. DOI: 10.1134/S0001437008030168.

4) Paragraph 2.2.3: indicate the scale of the pH measurement and again the temperature. The international community working on $CO_2$ fluxes, ocean acidification and impacts, has decided to adopt common protocols and common measurement units in order to increase utilization of data among different scientific communities. This uniformity would increase a wider utility of the paper. Protocols reported by Dickson et al 2007 that authors cite, are recommended. Please refer to them for units and scale. Line 140-141: authors should provide the accuracy of the method, for consistency with TA. This can be done by calibration against the reference materials (CRM's supplied by Dickson) and using CO2SYS for calculating the $pH_T$ of CRMs at the temperature of analysis. Specific comments at point 3 and 4 are necessary also for the next paragraph (2.2.4),

where CO2SYS programme is mentioned. This could be useful to non expert (of carbonate system analysis) readers.

The text will be re-written as:

"2.2.3 pH

A potentiometric method was applied to determine pH in the Pitzer pH scale (Pitzer, 1991) using a closed cell thermostated at 20°C with a sodium and hydrogen glass electrode pair without liquid junctions (Tishchenko et al., 2001, 2011). The buffer solution TRIS–TRIS–HCl– NaCl–H$_2$O (Tishchenko, 2000a) was used for calibrations in the Pitzer pH scale. Using this buffer not only the hydrogen glass electrode but also the sodium glass electrode was calibrated. Together with thermodynamic data (Dickson, 1990) the pH values were converted from the Pitzer pH scale to the total hydrogen ion concentration scale (Dickson et al., 2007). The accuracy of pH measurements was about 0.004 pH units".

Note, that pH values were measured potentiometrically in the Pitzer pH scale and reported at total scale according to method, developed by Prof. Pavel Ya. Tishchenko, the contributor of "Guide to Best Practices for Ocean CO$_2$ Measurements", edited by A.G. Dickson, C.L. Sabine, J.R. Chistain (2007). Direct comparison between these potentiometric and spectrophotometric pH values (both in "total" scale) demonstrated a good coincidence (Tishchenko et al., 2001). More details can be found in (Tishchenko et al., 2000ab, 2001, 2002, 2011).

Dickson, A.G. (1990). Standard potential of the reaction: AgCl(s) + 1/2 H2(g) = Ag(s) + HCl(aq), and the standard acidity constant of the ion HSO4$^-$ in synthetic sea water from 273.15 to 318.15 K, Journal of Chemical Thermodynamics, 22, 113-127.

Pitzer, K.S. (1991). Ionic interaction approach: theory and data correlation. In: Pitzer, K.S. (Ed.), Activity Coefficients in Electrolyte Solutions second ed. CRC Press, London, pp. 75–153.

Tishchenko, P.Ya. (2000a). Non-ideal properties of the TRIS–TRIS - HCl–NaCl–H2O buffer system in the 0–40 °C temperature interval. Application of the Pitzer equations, Izv. Akad. Nayk. Ser. Khim., 49, 670–675 (in Russian).

Tishchenko, P.Ya. (2000b). Standardization of pH measurements based on the ionic interaction approach, Izv. Akad. Nayk. Ser. Khim., 49,676–680 (in Russian).

Tishchenko, P.Ya., Wong, C.S., Pavlova, G.Yu, Johnson, W.K., Kang, D.-J., and Kim, K.-R. (2001). pH measurements of seawater by means of cell without liquid junction. Oceanology, 41, 6, 813–822.

Tishchenko, P.Ya., Il'ina, E.M., Chichkin, R.V., and Wong, C.S. (2002). pH measurements in estuary by means of cell without liquid junction. Oceanology, 42, 1, 27–35.

Tishchenko P. Ya., Kang D.-J., Chichkin R.V., Lazaryuk A.Yu., Wong C. S., Johnson W. K. (2011). Application of potentiometric method using a cell without liquid junction to underway pH measurements in surface seawater, Deep-Sea Research I, 58, 778–786.

5) Paragraph 2.2.4, lines 148-149: in order to prevent misundersting and not confuse direct continuous $pCO_2$ measurements (by SAMI CO2 sensor) with the calculated $pCO_2$ from discrete samples (collected by Rosette), I suggest to specify "At oceanographic stations surface $pCO_2$ values were calculated, on discrete samples, from $pH_T25$, AT and inorganic nutrients data using CO2SYS. . ." In addition authors should say which constants for sulfate and borate (KSO4 and KBorate) have been chosen in the CO2SYS programme.

It will be specified accordingly.

6) Lines 367-374: rephrase the two paragraphs as "In order to compare our estimates with those calculated by Lauvset et al. (2013) which carefully assessed the seasonal cycle of air-sea $CO_2$ fluxes in the Barents Sea, daily wind speed and quadratic parameterization of gas transfer velocity (Wanninkhof, 1992) were used for calculating CO2 fluxes in the northern Barents Sea. The $CO_2$ uptake during the 2007 fall season reached an average . . ... As the dataset by Lauvset et al. (2013) did not cover the north of the sea comprehensively, the data obtained during our cruise adds information enabling more accurate estimation of the absorption capacity of the whole Barents Sea in the fall season."

Thank you very much for suggestion, text will be replaced.

7) Lines 408-409: again I feel necessity of a clear indication that $pCO_2$ data of the selected transect, reported in fig 10, are calculated for discrete samples (from AT, $pH_T25$ and inorganic nutrients data) by means of CO2SYS programme.

It will be clarified in the text.

8) Fig 10 seems underutilized in the text, as only surface data are compared without any further discussion about vertical distributions. As the figure is very informative could you please comment a bit more ?

We have added a bit more details in describing the data of Figure 10. This text will be incorporated in the manuscript instead of the sentence on Line 410.

"The salinity distribution along the transect during the three cruises shows a similar general pattern, but with some significant variations especially in the top 30-50 m. Of the three years, 2007 had the lowest surface salinity and the most pronounced halocline (Figure 10). However, the largest interannual differences were in the seawater temperature distribution. In late summer of 2007 the surface layer was the warmest and underlain by a sharp thermocline coinciding in depth with the halocline to form a strong pycnocline that restricted vertical exchange. A

characteristic feature of the vertical distribution of $pCO_2$ over the transect in late summer 2007 was a pronounced subsurface maximum of $pCO_2$ (Figure 10) and higher $pCO_2$ values in the surface waters. Subsurface maximum was found exactly at the slope, and coincided with a layer of the brine-enriched south-eastern Laptev Sea bottom waters (Bauch et al., 2011). During years with prevalent offshore wind setting, such brine-enriched waters are exported to the Arctic Ocean halocline at about 50 m water depth (Bauch et al., 2009, 2011).

Westerly winds during the ice –free period in the summer 2007 advected the Lena River plume to the northeast. Thus, the low surface salinity was mainly related…" (Followed by text on Line 411).

The reference will be added in the manuscript:

Bauch, D., I. A. Dmitrenko, C. Wegner, J. Ho¨lemann, S. A. Kirillov, L. A. Timokhov, and Kassens H. (2009). Exchange of Laptev Sea and Arctic Ocean halocline waters in response to atmospheric forcing, J. Geophys. Res., 114, C05008, doi:10.1029/2008JC005062.

9) Line 421: I find a bit "dangerous" using here the word "supersaturation" as this make me to wonder if supersaturation has been really computed

Actually, it was "a weak supersaturation" in the south of the transect. It will be re-written.

10) Line 422: I find not fully proper to say that CO2 outgassing into the atmosphere was observed . . .. (Fig 10), as the calculated fluxes for the Laptev and East Siberian seas were really very low (see Tab 1). I would prefer rephrase as "Thus $\Delta pCO2$ conditions (Tab 1) favoring $CO_2$ outgassing into the atmosphere were observed"

Thank you. It will be changed accordingly.

11) Line 456-458: I suggest authors to rephrase as ". . . resulting in an increase of the area where seawater $pCO_2$ was in equilibrium with atmosphere and consequent reduction of $CO_2$ adsorption in the East Siberian Arctic seas".

It will be done.